# Stability Analysis and Generalization Bounds of Adversarial Training

**Jiancong Xiao[1,*], Yanbo Fan[2,†], Ruoyu Sun[1,3,†], Jue Wang[2], Zhi-Quan Luo[1,3]**
[1]The Chinese University of Hong Kong, Shenzhen;
[2]Tencent AI Lab; [3]Shenzhen Research Institute of Big Data
jiancongxiao@link.cuhk.edu.cn, fanyanbo0124@gmail.com,
sunruoyu@cuhk.edu.cn, arphid@gmail.com, luozq@cuhk.edu.cn

## Abstract

In adversarial machine learning, deep neural networks can fit the adversarial examples on the training dataset but have poor generalization ability on the test set. This phenomenon is called robust overfitting, and it can be observed when adversarially training neural nets on common datasets, including SVHN, CIFAR-10, CIFAR-100, and ImageNet. In this paper, we study the robust overfitting issue of adversarial training by using tools from uniform stability. One major challenge is that the outer function (as a maximization of the inner function) is nonsmooth, so the standard technique (*e.g.,* (Hardt et al., 2016)) cannot be applied. Our approach is to consider $\eta$-approximate smoothness: we show that the outer function satisfies this modified smoothness assumption with $\eta$ being a constant related to the adversarial perturbation $\epsilon$. Based on this, we derive stability-based generalization bounds for stochastic gradient descent (SGD) on the general class of $\eta$-approximate smooth functions, which covers the adversarial loss. Our results suggest that robust test accuracy decreases in $\epsilon$ when $T$ is large, with a speed between $\Omega(\epsilon\sqrt{T})$ and $\mathcal{O}(\epsilon T)$. This phenomenon is also observed in practice. Additionally, we show that a few popular techniques for adversarial training (*e.g.,* early stopping, cyclic learning rate, and stochastic weight averaging) are stability-promoting in theory.

## 1 Introduction

Deep neural networks (DNNs) (Krizhevsky et al., 2012; Hochreiter and Schmidhuber, 1997) have become successful in many machine learning tasks and rarely suffered overfitting issues (Zhang et al., 2021). A neural network model can be trained to achieve zero training error and generalize well to the unseen data. While in the setting of adversarial training, robust overfitting is a dominant issue (Rice et al., 2020). Specifically, robust overfitting characterizes a training procedure shown in Fig. 1. After a particular epoch, the robust test accuracy (black line) starts to decrease, but the robust training accuracy (blue line) is still increasing. This phenomenon can be observed in the experiments on common datasets, *e.g.,* SVHN, CIFAR-10, CIFAR-100, and ImageNet. Recent works (Gowal et al., 2020; Rebuffi et al., 2021) mitigated the overfitting issue using regularization techniques such as stochastic weight averaging (SWA)

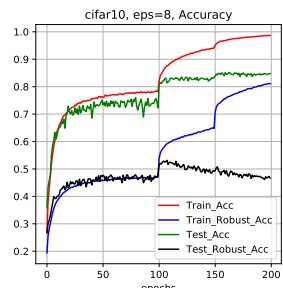

Figure 1: Experiments of adversarial training on CIFAR-10.

---

[*]This work is done when Jiancong Xiao is a research intern at Tencent AI Lab, China.
[†]Corresponding Authors.

36th Conference on Neural Information Processing Systems (NeurIPS 2022).

and early stopping, but it still has a large generalization gap. Therefore, it is essential to study this issue from a theoretical perspective. In this paper, we study the robust overfitting issue of adversarial training by using tools from uniform stability.

Uniform stability analysis (Bousquet and Elisseeff, 2002) in learning problems has been introduced to measure generalization gap instead of uniform convergence analysis such as classical VC-dimension (Vapnik and Chervonenkis, 2015) and Rademacher complexity (Bartlett and Mendelson, 2002). Generalization gap can be bounded in terms of uniform argument stability (UAS). Formally, UAS is the gap between the output parameters $\theta$ of running an algorithm $A$ on two datasets $S$ and $S'$ differ in at most one sample, denoted as $\delta(S, S') = \|\theta(S) - \theta(S')\|$. In a standard training problem with $n$ training samples, assuming that the loss function is convex, $L$-Lipschitz and $\beta$-gradient Lipschitz, running stochastic gradient descent (SGD) with step size $\alpha \leq 1/\beta$ for $T$ steps, the UAS is bounded by $\mathcal{O}(L\alpha T/n)$ (Hardt et al., 2016). Since the generalization gap is controlled by the number of samples $n$, this bound (at least partially) explains the good generalization of a standard training problem. Beyond (Hardt et al., 2016), Bassily et al. (2020) considered the case that the loss function is non-smooth. Without the $\beta$-gradient Lipschitz assumption, they showed that the UAS is bounded by $\mathcal{O}(L\alpha\sqrt{T} + L\alpha T/n)$ and provided a lower bound to show that the additional term $\mathcal{O}(L\alpha\sqrt{T})$ is unavoidable. In adversarial settings, two works have discussed the stability of adversarial training to our knowledge.

Firstly, Farnia and Ozdaglar (2021) considered the stability of minimax problems. Their work includes a discussion on an algorithm called GDmax (gradient descent on the maximization of the inner problem), which can be viewed as a general form of adversarial training. It provides the stability of GDmax when the inner problem is further assumed to be strongly concave. Under the strongly concave assumption, the outer function is smooth. Then, the generalization bound $\mathcal{O}(L\alpha T/n)$ can be applied in this case. However, the inner problem is not strongly concave in practice. The bound $\mathcal{O}(L\alpha T/n)$ does not match the poor generalization of adversarial training observed in practice.

Another stability analysis of adversarial training is the work of (Xing et al., 2021a). Before they propose their algorithm, they use the stability bound $\mathcal{O}(L\alpha\sqrt{T} + L\alpha T/n)$ (Bassily et al., 2020) to characterize the issue of robust generalization. However, the bound is $\epsilon$-independent. Let us Consider two cases, $\epsilon \to 0$ and $\epsilon$ is large (*e.g.,* 16/255). In the first case, adversarial training is very close to standard training and has good generalization ability. In the second case, robust generalization is very poor. The bound is the same in these two different cases. Therefore, it cannot provide an interpretation of robust generalization.

In summary, existing stability-based generalization bounds (Hardt et al., 2016; Bassily et al., 2020) have limited interpretation on robust generalization. In this paper, we provide a new stability analysis of adversarial training using the notion of $\eta$-approximate smoothness. We first show that, under the same assumptions as (Hardt et al., 2016; Xing et al., 2021a), even though the outer function (adversarial loss) is non-smooth, it is $\eta$-approximately smooth (see Definition 4.1), where $\eta$ is a constant linearly depending on the gradient Lipschitz of the inner function and the perturbation intensity $\epsilon$. Then, we derive stability-based generalization bounds (Thm. 5.1 to 5.4) for stochastic gradient descent (SGD) on this general class of $\eta$-approximate smooth functions, which covers the adversarial loss. Our main result can be summarized in the following equation. Running SGD on adversarial loss for $T$ steps with step size $\alpha \leq 1/\beta$, the excess risk, which is the sum of generalization error and optimization error, satisfies

$$\text{Excess Risk} \leq \mathcal{E}_{gen} + \mathcal{E}_{opt} \leq \overbrace{L\eta T\alpha}^{\text{additional}} + \overbrace{\frac{2L^2 T\alpha}{n} + \frac{D^2}{T\alpha} + L^2\alpha}^{\text{for standard training}}. \tag{1.1}$$

$$\underbrace{\phantom{L\eta T\alpha + \frac{2L^2 T\alpha}{n} + \frac{D^2}{T\alpha} + L^2\alpha}}_{\text{for adversarial training}}$$

The excess risk of adversarial training has an additional term $L\eta T\alpha$. We also provide a lower bound of UAS of running SGD on adversarial loss. Our results suggest that robust test accuracy decreases in $\epsilon$ when $T$ is large, with a speed between $\Omega(\epsilon\sqrt{T})$ and $\mathcal{O}(\epsilon T)$. This phenomenon is also observed in practice. It provides an understanding of robust overfitting from the perspective of uniform stability. Additionally, we show that a few popular techniques for adversarial training (*e.g.,* early stopping, cyclic learning rate, and stochastic weight averaging) are stability-promoting in theory and also empirically improve adversarial training. Experiments on SVHN, CIFAR-10, CIFAR-100, and ImageNet confirm our results. Our contributions are listed as follows:

- Main results: we derive stability-based generalization bounds for adversarial training using the notion of $\eta$-approximate smoothness. Based on this, we provide an analysis to understand robust overfitting.
- We provide the stability analysis of a few popular techniques for adversarial training and show that they are indeed stability-promoting.
- We provide experiments on SVHN, CIFAR-10, CIFAR-100, and ImageNet. The results verify the generalization bounds.
- Technical contribution: we develop a set of properties of $\eta$-approximately smooth function, which might be useful in other tasks.

The paper is organized as follows. After discussing the related work in Sec. 2, the rest of the paper contains two parts. The first part is the technical part to derive stability bounds. The second part is to analyze robust overfitting. Specifically, in the first part, Sec. 3 introduces the preliminary knowledge about UAS. Sec. 4 provides the Lemma and properties of approximately smooth functions and Sec. 5 gives the stability bounds. In the second part, Sec. 6 analyzes the robust overfitting in the theoretical settings and Sec. 7 presents the experiments.

## 2 Related Work

**Adversarial Attacks and Defense.** Starting from the work of (Szegedy et al., 2013), it has been commonly realized that deep neural networks are highly susceptible to imperceptible corruptions to the input data (Goodfellow et al., 2014; Carlini and Wagner, 2017; Madry et al., 2017). A series of work aimed at training neural networks robust to such small perturbations (Wu et al., 2020; Gowal et al., 2020; Zhang et al., 2020) and another line of work aimed at designing more powerful adversarial attack algorithms (Athalye et al., 2018; Tramer et al., 2020; Fan et al., 2020; Xiao et al., 2022c; Qin et al., 2022). A series of work considered adversarial robustness in black-box settings (Chen et al., 2017; Qin et al., 2021). Semi-supervised learning has been used to improve adversarial robustness (Carmon et al., 2019; Li et al., 2022). Fast adversarial training (Wong et al., 2020; Huang et al., 2022) was introduced to save training time.

**Robust Generalization.** A series of work tried to explain robust generalization in the uniform convergence framework, including VC-dimension (Attias et al., 2021; Montasser et al., 2019) and Rademacher complexity (Khim and Loh, 2018; Yin et al., 2019; Awasthi et al., 2020; Xiao et al., 2022a). Uniform algorithmic stability is another framework to study robust generalization (Farnia and Ozdaglar, 2021; Xing et al., 2021a; Xiao et al., 2022b). The work of (Schmidt et al., 2018; Raghunathan et al., 2019; Zhai et al., 2019) have shown that in some scenarios achieving robust generalization requires more data. The work of (Xing et al., 2021b,c; Javanmard et al., 2020) studied the generalization in the setting of adversarial linear regression. (Sinha et al., 2017) studied the generalization of distributional robustness. The work of (Taheri et al., 2020; Javanmard et al., 2020; Dan et al., 2020) analyzed robust generalization in Gaussian mixture models.

**Uniform Stability.** Stability is a classical approach to provide generalization bounds. It can be traced back to the work of (Rogers and Wagner, 1978). After a few decades, it was well developed in analyzing the generalization bounds in statistical learning problems (Bousquet and Elisseeff, 2002). These bounds have been significantly improved in a recent sequence of works (Feldman and Vondrak, 2018, 2019). The work of (Chen et al., 2018) derived minimax lower bounds for excess risk and discussed the optimal trade-off between stability and convergence. Ozdaglar et al. (2022) considered the generalization metric of minimax optimizer.

## 3 Preliminaries of Stability

Consider the following setting of statistical learning. There is an unknown distribution $\mathcal{D}$ over examples from some space $\mathcal{Z}$. We receive a sample dataset $S = \{z_1, \ldots, z_n\}$ of $n$ examples drawn i.i.d. from $\mathcal{D}$. The *population risk* and *empirical risk* are defined as:

$$R_{\mathcal{D}}(\theta) \overset{\text{def}}{=} \mathbb{E}_{z \sim \mathcal{D}}\, h(\theta, z) \quad \text{and} \quad R_S(\theta) \overset{\text{def}}{=} \frac{1}{n} \sum_{i=1}^{n} h(\theta, z_i),$$

respectively, where $h(\cdot, \cdot)$ is the loss function.

**Risk Decomposition.** Let $\theta^*$ and $\bar{\theta}$ be the optimal solution of $R_{\mathcal{D}}(\theta)$ and $R_S(\theta)$ respectively. Then for the algorithm output $\hat{\theta} = A(S)$, the excess risk can be decomposed as

$$R_{\mathcal{D}}(\hat{\theta}) - R_{\mathcal{D}}(\theta^*) = \underbrace{R_{\mathcal{D}}(\hat{\theta}) - R_S(\hat{\theta})}_{\mathcal{E}_{gen}} + \underbrace{R_S(\hat{\theta}) - R_S(\bar{\theta})}_{\mathcal{E}_{opt}} + \underbrace{R_S(\bar{\theta}) - R_S(\theta^*)}_{\leq 0} + \underbrace{R_S(\theta^*) - R_{\mathcal{D}}(\theta^*)}_{\mathbb{E} = 0}.$$

To control the excess risk, we need to control the generalization gap $\mathcal{E}_{gen}$ and the optimization gap $\mathcal{E}_{opt}$. In the rest of the paper, we use $\mathcal{E}_{gen}$ and $\mathcal{E}_{opt}$ to denote the *expectation* of the generalization and optimization gap. To bound the generalization gap of a model $\hat{\theta} = A(S)$ trained by a randomized algorithm $A$, we employ the following notion of *uniform stability*.

**Definition 3.1.** *A randomized algorithm $A$ is $\varepsilon$-uniformly stable if for all data sets $S, S' \in \mathcal{Z}^n$ such that $S$ and $S'$ differ in at most one example, we have*

$$\sup_z \mathbb{E}_A \left[ h(A(S); z) - h(A(S'); z) \right] \leq \varepsilon. \tag{3.1}$$

Here, the expectation is taken only over the internal randomness of $A$. We recall the important theorem that uniform stability implies *generalization in expectation* (Hardt et al., 2016).

**Theorem 3.1** (Generalization in expectation). *Let $A$ be $\varepsilon$-uniformly stable. Then, the expected generalization gap satisfies*

$$|\mathcal{E}_{gen}| = |\mathbb{E}_{S,A}[R_{\mathcal{D}}[A(S)] - R_S[A(S)]]| \leq \varepsilon.$$

Therefore, we turn to the properties of iterative algorithms that control their uniform stability.

## 4 Stability of Adversarial Training

**Adversarial Surrogate Loss.** In adversarial training, we consider the following surrogate loss

$$h(\theta; z) = \max_{\|z - z'\|_p \leq \epsilon} g(\theta; z'), \tag{4.1}$$

where $g(\theta; z)$ is the loss function of the standard counterpart, $\|\cdot\|_p$ is the $\ell_p$-norm, $p \geq 1$. Usually, $g$ can also be written in the form of $\ell(f_\theta(\boldsymbol{x}); y)$, where $f_\theta$ is the neural network to be trained and $(\boldsymbol{x}, y)$ is the input-label pair. We assume the loss function $g$ satisfies the following smoothness assumption.

**Assumption 4.1.** *The function $g$ satisfies the following Lipschitzian smoothness conditions:*

$$\|g(\theta_1, z) - g(\theta_2, z)\| \leq L\|\theta_1 - \theta_2\|,$$
$$\|\nabla_\theta g(\theta_1, z) - \nabla_\theta g(\theta_2, z)\| \leq L_\theta\|\theta_1 - \theta_2\|,$$
$$\|\nabla_\theta g(\theta, z_1) - \nabla_\theta g(\theta, z_2)\| \leq L_z\|z_1 - z_2\|_p,$$

*where $\|\cdot\|$ is Euclidean norm.*

Assumption 4.1 assumes that the loss function is smooth, which are also used in the stability literature (Farnia and Ozdaglar, 2021; Xing et al., 2021a), as well as the convergence analysis literature (Wang et al., 2019; Liu et al., 2020). While ReLU activation function is non-smooth, recent works (Allen-Zhu et al., 2019; Du et al., 2019) showed that the loss function of over-parameterized DNNs is semi-smooth. It helps justify Assumption 4.1. Under Assumption 4.1, the loss function of adversarial training satisfies the following Lemma (Liu et al., 2020).

**Lemma 4.1.** *Let $h$ be the adversarial loss defined in Eq. (4.1) and $g$ satisfies Assumption 4.1. $\forall \theta_1, \theta_2$ and $\forall z \in \mathcal{Z}$, the following properties hold.*

1. *(Lipschitz function.) $\|h(\theta_1, z) - h(\theta_2, z)\| \leq L\|\theta_1 - \theta_2\|$.*

2. *For all subgradient $d(\theta, z) \in \partial_\theta h(\theta, z)$, we have $\|d(\theta_1, z) - d(\theta_2, z)\| \leq L_\theta\|\theta_1 - \theta_2\| + 2L_z\epsilon$.*

If we further assume that $g(\theta, z)$ is $\mu$-strongly concave in $z$, the adversarial surrogate loss $h(\theta, z)$ is also smooth (Sinha et al., 2017). Therefore, the uniform stability of adversarial training follows (Hardt et al., 2016; Farnia and Ozdaglar, 2021), see Appendix B.3. However, $g(\theta, z)$ is non-strongly concave in practice. The above results provide limited explanations of the poor generalization of adversarial training. We discuss the generalization properties of adversarial training under Lemma 4.1.2.

## 4.1 Basic Properties of Approximate Smoothness

To simplify the notation, we use $h(\theta)$ as a shorthand notation of $h(\theta, z)$. To simplify the argument, we consider differentiable function $h$. The results can be extended to non-differentiable cases. Lemma 4.1 motivates us to analyze a function with the following modified smoothness assumption, which we call approximate smoothness assumption.

**Definition 4.1.** *Let $\beta > 0$ and $\eta > 0$. We say a differentiable function $h(\theta)$ is $\eta$-approximately $\beta$-gradient Lipschitz, if $\forall \theta_1$ and $\theta_2$, we have*

$$\|\nabla h(\theta_1) - \nabla h(\theta_2)\| \le \beta\|\theta_1 - \theta_2\| + \eta.$$

In definition 4.1, $\eta$ controls the smoothness of the loss function $h(\cdot)$. When $\eta = 0$, the function $h$ is gradient Lipschitz. When $\eta \to +\infty$, $h$ is a general non-smooth function. As our discussion before, the adversarial surrogate loss is $2L_z\epsilon$-approximately smooth. As far as we know, this assumption is rarely discussed in the optimization literature since it cannot improve the convergence rate from a general non-smooth assumption. But it affects uniform stability, as we will discuss later. We need to develop the basic properties of approximate smoothness first.

**Lemma 4.2.** *Assume that the function $h$ is $\eta$-approximately $\beta$-gradient Lipschitz. $\forall \theta_1, \theta_2$ and $\forall z \in \mathcal{Z}$, the following properties hold.*

1. *($\eta$-approximate descent lemma.)*

$$h(\theta_1) - h(\theta_2) \le \nabla h(\theta_2)^T(\theta_1 - \theta_2) + \frac{\beta}{2}\|\theta_1 - \theta_2\|^2 + \eta\|\theta_1 - \theta_2\|.$$

2. *($\eta$-approximately co-coercive.) Assume in addition that $h(\theta, z)$ is convex in $\theta$ for all $z \in \mathcal{Z}$. Let $[\cdot]_+ = \max(0, \cdot)$. We have*

$$\langle \nabla h(\theta_1) - \nabla h(\theta_2), \theta_1 - \theta_2 \rangle \ge \frac{1}{\beta}\left[[\|\nabla h(\theta_1) - \nabla h(\theta_2)\| - \eta]_+\right]^2.$$

We defer the proof to Appendix A. Note that the loss function is $L$-Lipschitz for every example $z$, we have $\mathbb{E}|h(\theta_1, z) - h(\theta_2, z)| \le L\mathbb{E}\|\theta_1 - \theta_2\|$, for all $z \in \mathcal{Z}$. To obtain the stability generalization bounds, we need to analysis the difference $\|\theta_1^T - \theta_2^T\|$, where $\theta_1^T$ and $\theta_2^T$ are the outputs of running SGD on adversarial surrogate loss for $T$ iterations on two datasets with only one different sample. Next we provide the recursive bounds under the approximate smoothness assumption.

**Algorithms.** We consider the stochastic gradient descent on the adversarial surrogate loss. i.e.,

$$\theta^{t+1} = \theta^t - \alpha_t \nabla_\theta h(\theta^t, z_{i_t}), \tag{4.2}$$

where $\alpha_t$ is the step size in iteration $t$, $z_{i_t}$ is the sample chosen in iteration $t$. We consider two popular schemes for choosing the examples indices $i_t$. *Sampling with replacement*: One is to pick $i_t \sim \text{Uniform}\{1, \cdots, n\}$ at each step. *Fixed permutation*: The other is to choose a random permutation over $\{1, \cdots, n\}$ and cycle through the examples repeatedly in the order determined by the permutation. Our results hold for both variants.

**Properties of Update Rules.** We define $G_{\alpha,z}(\theta) = \theta - \alpha\nabla h(\theta, z)$ be the update rule of SGD. The following lemma holds.

**Lemma 4.3.** *Assume that the function $h$ is $\eta$-approximately $\beta$-gradient Lipschitz. $\forall \theta_1, \theta_2$ and $\forall z \in \mathcal{Z}$, we have*

1. *($\alpha\eta$-approximately $(1 + \alpha\beta)$-expansive.) $\|G_{\alpha,z}(\theta_1) - G_{\alpha,z}(\theta_2)\| \le (1 + \alpha\beta)\|\theta_1 - \theta_2\| + \alpha\eta$.*

2. *($\alpha\eta$-approximately non-expansive.) Assume in addition that $h(\theta, z)$ is convex in $\theta$ for all $z \in \mathcal{Z}$, for $\alpha \le 1/\beta$, we have $\|G_{\alpha,z}(\theta_1) - G_{\alpha,z}(\theta_2)\| \le \|\theta_1 - \theta_2\| + \alpha\eta$.*

3. *($\alpha\eta$-approximately $(1 - \alpha\gamma)$-contractive.) Assume in addition that $h(\theta, z)$ is $\gamma$-strongly convex in $\theta$ for all $z \in \mathcal{Z}$, for $\alpha \le 1/\beta$, we have $\|G_{\alpha,z}(\theta_1) - G_{\alpha,z}(\theta_2)\| \le (1 - \alpha\gamma)\|\theta_1 - \theta_2\| + \alpha\eta$.*

The proof of Lemma 4.3 is based on Lemma 4.2 and is deferred to Appendix A. Lemma 4.3 provides the recursive distance $\|\theta_1 - \theta_2\|$ from iteration $t$ to $t + 1$. Based on this, we can recursively derive the distance $\|\theta_1^T - \theta_2^T\|$. Then, we can obtain the stability generalization bounds.

# 5 Stability Generalization Bounds

In the previous section, we have discussed the properties of approximate smoothness and developed tools we need to use. In this section, we discuss the stability bounds.

## 5.1 Convex Optimization

We first consider the case that $h(\theta, z)$ is convex in $\theta$ for all $z \in \mathcal{Z}$.

**Theorem 5.1.** *Assume that $h(\theta, z)$ is convex, $L$-Lipschitz, and $\eta$-approximately $\beta$-gradient Lipschitz in $\theta$ for all given $z \in \mathcal{Z}$. Suppose that we run SGD with step sizes $\alpha_t \leq 1/\beta$ for $T$ steps. Then,*

$$\mathcal{E}_{gen} = \mathbb{E}[R_{\mathcal{D}}(\theta^T) - R_S(\theta^T)] \leq L\left(\eta + \frac{2L}{n}\right) \sum_{t=1}^{T} \alpha_t. \tag{5.1}$$

The proof is mainly based on Lemma 4.3 that the update rule is approximately non-expansive in this case. We defer it to appendix A. We have shown that the adversarial surrogate loss is $2L_z\epsilon$-approximately $L_\theta$-gradient Lipschitz. Let $\eta = 2L_z\epsilon$ in Eq. (5.1), we directly obtain the stability bounds for adversarial training. In practice, the solution of the inner problem is sub-optimal. Let $\Delta\epsilon$ be the maximum error between the optimal and sub-optimal attacks in each iteration. We have the following Corollary.

**Corollary 5.1** (Uniform stability for sub-optimal attacks adversarial training.). *Under Assumption 4.1, assume in addition that $g(\theta, z)$ is convex in $\theta$ for all given $z \in \mathcal{Z}$. Suppose that we run adversarial training with step sizes $\alpha_t \leq 1/L_\theta$ for $T$ steps. Then, adversarial training satisfies uniform stability with*

$$\mathcal{E}_{gen} \leq \left(2LL_z(\epsilon + \Delta\epsilon) + \frac{2L^2}{n}\right) \sum_{t=1}^{T} \alpha_t \leq \mathcal{O}\left(L(L_z\epsilon + \frac{L}{n}) \sum_{t=1}^{T} \alpha_t\right). \tag{5.2}$$

**Remark:** Corollary 5.1 shows that adversarial training with weak attacks (large $\Delta\epsilon$) have worse robust generalization than that with strong attacks. This is also observed in practice. However, $\Delta\epsilon$ is at most $2\epsilon$, the upper bound of AT with different attacks have the same order. It might be due to the weakness of Assumption 4.1 or uniform stabilty framework.

**Interpreting Robust Generalization.**   If we let $\epsilon = 0$ in Eq. (5.2), it reduced to the generalization bound in (Hardt et al., 2016) for standard training. Therefore, the additional generalization error of adversarial training comes from the additional term $L_z\epsilon$. The global gradient Lipschitz $L_z$ with respect to $z$ plays an important role in generalization. Even though the perturbation $\epsilon$ is small, it is amplified by the Lipschitz $L_z$ and finally hurts the robust generalization.

## 5.2 Further Discussion on the Generalization Bounds

We first provide a lower bound. Then we compare our bounds with the existing bounds in Table 1.

**Theorem 5.2** (Lower Bound). *There exists functions $h(\theta, z)$, s.t. $h$ is convex, $L$-Lipschitz, and $\eta$-approximately $\beta$-gradient Lipschitz in $\theta$ for all given $z \in \mathcal{Z}$. Exists $S$ and $S'$ differ in one sample. Suppose that we run SGD with step fixed step sizes $\alpha \leq 1/\beta$ for $T$ steps. Then,*

$$\mathbb{E}[\delta(S, S')] \geq \Omega\left(\eta\alpha\sqrt{T} + \frac{L\alpha T}{n}\right). \tag{5.3}$$

Table 1: Comparison of the upper and lower bounds of $\mathbb{E}[\delta(S, S')]$. Comparing with the previous results, we replace $L$ by $\eta$ and provide the matching lower bound in $\eta$.

|  | Assumption | Upper bounds | Lower bounds |
|---|---|---|---|
| Farnia and Ozdaglar (2021) | convex-strongly concave | $\mathcal{O}(LT\alpha/n)$ | $\Omega(LT\alpha/n)$ |
| Xing et al. (2021a) | convex-nonconcave | $\mathcal{O}(L\sqrt{T}\alpha + LT\alpha/n)$ | $\Omega(\sqrt{T}\alpha + LT\alpha/n)$ |
| Ours | convex-nonconcave | $\mathcal{O}(\eta T\alpha + LT\alpha/n)$ | $\Omega(\eta\sqrt{T}\alpha + LT\alpha/n)$ |

**Comparison with the Existing UAS Bounds.** Compared with the work of (Farnia and Ozdaglar, 2021), they assume that the inner problem is strongly concave. Thus the bounds are not comparable.

Strongly concave is a strong assumption in practice. Therefore, the work of (Xing et al., 2021a) and our analysis focus on the nonconcave cases. Comparing with the bound $\mathcal{O}(L\sqrt{T}\alpha + LT\alpha/n)$, our bound captures a critical aspect of robust generalization bound: $\epsilon$-dependent. As observed in practice, the robust generalization gap reduces to the standard generalization gap as $\epsilon \to 0$. Our bound consists with this observation. On the contrary, the bound $\mathcal{O}(L\sqrt{T}\alpha + LT\alpha/n)$ is very large when $\epsilon \to 0$. Additionally, we provide a matching lower bound w.r.t $\eta$. The comparison is provided in Table 1.

**Comparison with the Work of (Xing et al., 2021a).** The work of (Xing et al., 2021a) argued that the max function is not smooth even though the standard counterpart is smooth. Therefore, they followed the bound in non-smooth cases (Bassily et al., 2020). Then, they aimed to solve the non-smooth issue. They design a noise-injected algorithm and show its effectiveness in tackling the non-smooth issue. Our work focus on providing better bounds to interpret robust overfitting.

## 5.3 Non-convex Optimization and Strongly Convex Optimization

Next, we consider the case that the loss function $h$ is general non-convex and strongly convex. By Lemma 4.3, we have

**Theorem 5.3.** *Assume that $h(\theta, z)$ is $L$-Lipschitz, and $\eta$-approximately $\beta$-gradient Lipschitz in $\theta$ for all given $z \in \mathcal{Z}$. Assume in addition that $0 \le h(\theta, z) \le B$ for all $\theta$ and $z$. Suppose that we run SGD with diminishing step sizes $\alpha_t \le 1/(\beta t)$ for $T$ steps. Then*

$$\mathcal{E}_{gen} \le \frac{BL_\theta + (2L^2 + L\eta n)T}{\beta(n-1)}. \tag{5.4}$$

**Theorem 5.4.** *Assume that $h(\theta, z)$ is $\gamma$-strongly convex, $L$-Lipschitz, and $\eta$-approximately $\beta$-gradient Lipschitz in $\theta$ for all given $z \in \mathcal{Z}$. Suppose that we run SGD with step sizes $\alpha_t \le 1/\beta$ for $T$ steps. Then*

$$\mathcal{E}_{gen} = \mathbb{E}[R_\mathcal{D}(\theta^T) - R_S(\theta^T)] \le \frac{L\eta}{\gamma} + \frac{2L^2}{\gamma n}. \tag{5.5}$$

**Remark:** The bound in non-convex cases provides a similar interpretation of robust generalization to the analysis in the convex case. In uniform stability analysis, whether the loss function is convex or non-convex does not give a major difference. Therefore, we provide the analysis of the non-convex case in Appendix B. We also provide our convergence analysis of running SGD on a $\eta$-approximate smoothness, non-convex function in Theorem B.1. Strongly convex is a strong assumption. We leave the analysis of the bound in strongly convex cases in Appendix B.2.

# 6 Excess Risk Minimization

Based on the risk decomposition, we have Excess Risk $\le \mathcal{E}_{gen} + \mathcal{E}_{opt}$, we need to minimize $\mathcal{E}_{gen} + \mathcal{E}_{opt}$ to achieve better performance. Per our previous discussion, whether the loss function is convex or non-convex does not give a major difference in stability analysis. We study the convex case in this section. We leave the discussion on the non-convex and strongly convex cases in Appendix B.1 and B.2, respectively. We first introduce the optimization error.

**Optimization Analysis.** The convergence analysis of SGD on a $L$-Lipschitz, convex function is discussed in (Nemirovski et al., 2009). The convergence rate cannot be improved if we further assume that the function $h$ is gradient Lipschitz. Therefore, a weaker condition, approximately gradient Lipschitz, cannot improve the convergence rate. We use the following convergence error bound (adopted from (Nemirovski et al., 2009)) for the optimization error of both adversarial training and standard training.

**Theorem 6.1.** *Assume that $h(\theta, z)$ is $L$-Lipschitz and convex in $\theta$ for all given $z \in \mathcal{Z}$. Let $D = \|\theta^0 - \theta^*\|$, where $\theta^0$ is the initialization of SGD. Suppose that we run SGD with step sizes $\alpha_t$ for $T$ steps. Then, $\exists k \le T$, s.t.*

$$\mathcal{E}_{opt}(\theta^k) \le \frac{D^2 + L^2 \sum_{t=1}^T \alpha_t^2}{\sum_{t=1}^T \alpha_t}. \tag{6.1}$$

If we let $\alpha_t = 1/\sqrt{T}$, we have $\mathcal{E}_{opt} \leq \mathcal{O}(1/\sqrt{T})$, which is the convergence rate of SGD on convex function. Since we need to consider the generalization and optimization errors simultaneously, we keep the $\alpha_t$ in Theorem 6.1.

**Generalization-Optimization Trade-off.** We have now discussed the optimization and generalization errors of adversarial training. We aim to find the optimal trade-off between generalization and optimization in terms of $\alpha_t$ and $T$. However, finding the optimal $\alpha_t$ and $T$ simultaneously is challenging. We consider the settings we use in practice.

**Fixed Step Size.** We first consider the simplest case, the step size $\alpha$ is fixed. Then, combining Eq. (5.1) and Eq. (6.1), we have

$$\mathcal{E}_{gen} + \mathcal{E}_{opt} \leq \underbrace{\overbrace{L\eta T\alpha}^{additional} + \overbrace{\frac{2L^2T\alpha}{n} + \frac{D^2}{T\alpha} + L^2\alpha}^{\text{for standard training}}}_{\text{for adversarial training}} . \tag{6.2}$$

**Interpretation of Robust Overfitting.** In standard training, overfitting is rarely observed in practice. The optimization error and generalization error are both small. In Eq. (6.2), the second to the fourth terms are for standard training. The second term is controlled by the number of samples $n$, which is small if we have sufficient training samples. The third term is controlled by $T$, and the last term is fixed given a small $\alpha$. This bound partially explains the good performance of standard training. However, we have an additional term $L\eta T\alpha$ for excess risk in adversarial training. Then, after a particular iteration that $L\eta T\alpha$ dominates Eq. (6.2), robust overfitting appears. This is consistent with the training procedure in practice. Therefore, the $\eta = 2L_z\epsilon$ approximate smoothness of adversarial loss provides a possible explanation of robust overfitting. To achieve better performance, we need to stop training the model earlier.

**Early Stopping.** It is shown that early stopping is an important training technique for adversarial training (Rice et al., 2020). In Eq. (6.2), if we optimize the right-hand-side with respect to $T$, we have

$$T^* = \frac{\|\theta^0 - \theta^*\|}{\alpha\sqrt{L\eta + 2L^2/n}}, \quad \mathcal{E}_{gen} + \mathcal{E}_{opt} \leq 2\sqrt{L\eta + \frac{2L^2}{n}}D + L^2\alpha.$$

Therefore, it is the best to stop training at $T^*$. However, $T^*$ is unknown in practice. It is important to select stopping criteria. For example, we can use a validation set to determine when to stop.

**Varying Step Size.** We discuss one popular varying step size schedule, cyclic learning rate, which is also called super-converge learning rate for adversarial training. It is shown that it can speed up adversarial training with fewer epochs (Wong et al., 2020). The Super-converge learning rate follows the following rules. In the first phase (warm-up), the step size increase from 0 to $\alpha'$ linearly. In the second phase (cold down), the step size decreases back to 0 linearly. It is unclear (to our knowledge) why this schedule can speed up convergence in optimization theory. But the generalization part can partially be explained by UAS. If we set $\alpha' = 2\alpha$, it is easy to check that $\mathcal{E}_{gen} \leq (L\eta + 2L^2/n)T\alpha$ in this case, which is the same as the bound in the fixed learning rate case. Notice that the cyclic learning rate usually requires fewer steps $T$ to converge. Then, the generalization gap is smaller. Cyclic learning rate can be viewed as another form of early stopping from the perspective of UAS.

**Stochastic Weight Averaging.** SWA is also a useful training technique for adversarial training (Hwang et al., 2021). Instead of using the last checkpoint, SWA suggests using the average of the checkpoints for inference. It is shown that SWA can find a model with better generalization since it leads to wider minima (Izmailov et al., 2018). Below we study SWA from the perspective of UAS.

**Theorem 6.2.** *Assume that $h(\theta, z)$ is convex, $L$-Lipschitz, and $\eta$-approximately $\beta$-gradient Lipschitz in $\theta$ for all given $z \in \mathcal{Z}$. Suppose that we run SGD with step sizes $\alpha_t \leq 1/\beta$ for $T$ steps. Let $\bar{\theta}$ be the average of the trajectory. Then,*

$$\mathcal{E}_{gen}(\bar{\theta}) \leq \left(\frac{L\eta}{2} + \frac{L^2}{n}\right)\sum_{t=1}^{T}\alpha_t, \quad \mathcal{E}_{opt}(\bar{\theta}) \leq \frac{\|\theta^0 - \theta^*\|^2 + L^2\sum_{t=1}^{T}\alpha_t^2}{\sum_{t=1}^{T}\alpha_t} . \tag{6.3}$$

In words, SWA reduces the generalization error bound to one-half of the one without SWA. But the training error bound remains unchanged.

## 7  Experiments

**Training Settings.**  We mainly consider the experiments on CIFAR-10 (Krizhevsky et al., 2009), CIFAR-100, and SVHN (Netzer et al., 2011). We also provide one experiment on ImageNet (Deng et al., 2009). For the first three datasets, we conduct the experiments on training PreActResNet-18, which follows (Rice et al., 2020), For the experiment on ImageNet, we use ResNet-50 (He et al., 2016), following the experiment of (Madry et al., 2017). For the inner problems, we adopt the $\ell_\infty$ PGD adversarial training in (Madry et al., 2017), the step size in the inner maximization is set to be $\epsilon/4$ on CIFAR-10 and CIFAR100 and is set to be $\epsilon/8$ on SVHN. Weight decay is set to be $5 \times 10^{-4}$. Additional experiments are provided in Appendix C. [3]

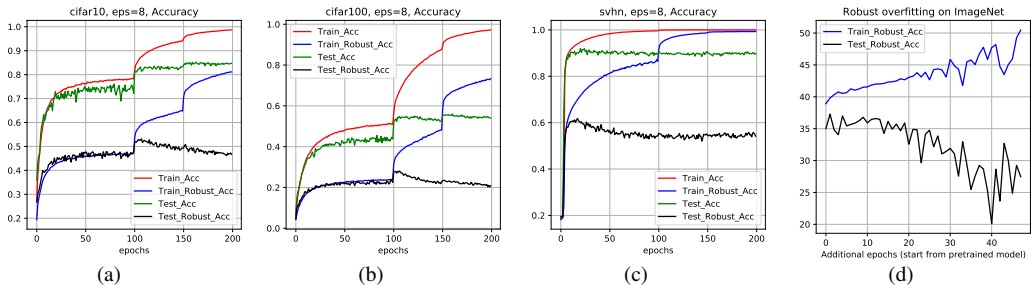

Figure 2: Robust overfitting in the experiments on (a) CIFAR-10, (b) CIFAR-100, (c) SVHN and (d) ImageNet.

**Robust Overfitting on Common Dataset.**  In Fig. 2 (a), (b), and (c), we show the experiments on the piece-wise learning rate schedule, which is 0.1 over the first 100 epochs, down to 0.01 over the following 50 epochs, and finally be 0.001 in the last 50 epochs, on CIFAR-10, CIFAR-100, and SVHN. Experiments on different $\epsilon$ are shown in Appendix C. *Robust Overfitting on ImageNet.* We provide one experiment on ImageNet in Fig. 2 (d). We start from a pre-trained model from Madry's Lab and keep running 50 more epochs. *Robust overfitting* can be observed in these experiments. After a particular epoch (around the $100^{th}$ epoch), the robust training accuracy is still increasing, but the robust test accuracy starts to decrease.

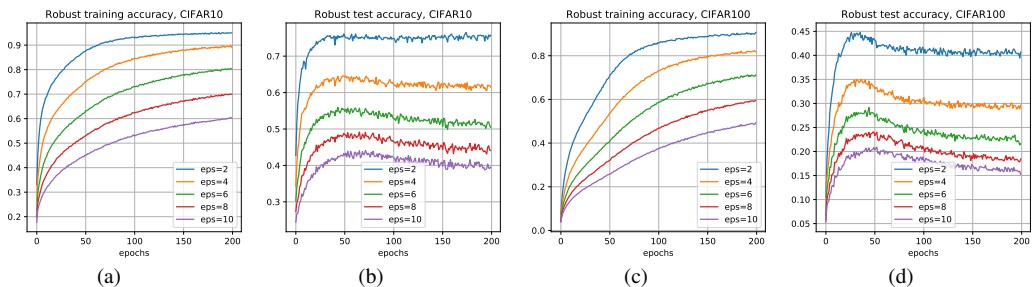

Figure 3: Experiments of adversarial training on CIFAR-10 and CIFAR-100 with a fixed learning rate. (a) Robust training accuracy on CIFAR-10. (b) Robust test accuracy on CIFAR-10. (c) Robust training accuracy on CIFAR-100. (d) Robust test accuracy on CIFAR-100. $\epsilon$ are set to be 2, 4, 6, 8, and 10.

---

[3]`https://github.com/JiancongXiao/Stability-of-Adversarial-Training`

**Fixed Step Size.** To better understand robust overfitting and match the theoretical settings (in Eq. (6.2)), we consider the fixed learning rate schedule. In Fig. 3, we show the experiments of adversarial training using a fixed learning rate $0.01$. The perturbation intensity $\epsilon$ is set to be 2, 4, 6, 8, and 10. respectively. Fig. 3 (a) and (b) show the experiments on CIFAR-10. Fig. 3 (c) and (d) show the experiments on CIFAR-100, respectively. Fig. 4 shows the experiments on SVHN.

**Generalization Error Dominates Training Error.** In the robust overfitting phase, the robust generalization error dominates the robust training error. This phenomenon corresponds to the Eq. (6.2) that the first term dominates the other terms with large $T$.

**Robust Test Accuracy Decreases in $\epsilon$.** Comparing different $\epsilon$ in Fig. 3, we can see that the robust test accuracy decreases faster when $\epsilon$ is larger. This corresponds to the robust overfitting rule in the theoretical setting that the test performance decreases in $\epsilon$. These phenomena are similar in theoretical and practical settings. Therefore, the stability analysis provides a different perspective on understanding robust overfitting.

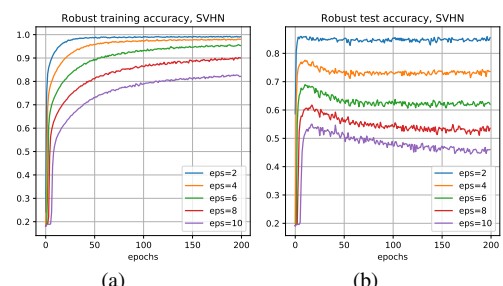

Figure 4: Experiments of adversarial training on SVHN with a fixed learning rate. (a) Robust training accuracy. (b) Robust test accuracy.

**Robust Test Accuracy Decreases in a Rate between $\Omega(\epsilon\sqrt{T})$ and $\mathcal{O}(\epsilon T)$.** If $\epsilon$ is small, *e.g.,* $\epsilon = 2$, the decrease rate is close to $\mathcal{O}(\epsilon T)$. If $\epsilon$ is large, the decrease rate is more likely to be $\Omega(\epsilon\sqrt{T})$. This is also the gap between the upper bound and lower bound in Sec. 5.

# 8 Conclusion

**Limitations and Future Work.** Firstly, in Fig. 3, we can see that the decrease rate of robust overfitting is close to $\mathcal{O}(T)$ when $\epsilon$ is small and is close to $\Omega(\sqrt{T})$ when $\epsilon$ is large. One possible direction is to figure out the relation. Secondly, one might improve adversarial training by controlling $L_z$. $L_z$ depends on the loss, the network architecture, and the dataset. One possible direction is to design a smoother loss or smooth activation function (e.g., SiLU) for a lower $L_z$. Notice that $L_z$ is uniform for all $\theta$. If we view $L_z(\theta)$ locally with respect to $\theta$, we might use an (approximated) second-order penalty term on its magnitude to control it.

In this paper, we show that the adversarial loss satisfies $\eta$-approximate smoothness, and we derive stability-based generalization bounds on this general class of $\eta$-approximate smooth functions. Our bounds give a different perspective on understanding robust overfitting. The robust test accuracy decreases in $\eta$, and experimental results confirm this phenomenon. We think our work will inspire more theoretical and empirical research to improve adversarial training.

# Acknowledgement

We would like to thank Jiawei Zhang, Congliang Chen, and Zeyu Qin for the helpful discussions. We thank all the anonymous reviewers for their comments and suggestions. The work is supported by NSFC-A10120170016, NSFC-617310018 and the Guangdong Provincial Key Laboratory of Big Data Computing.

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
