# A  Proof of the Theorem

## A.1  Proof of Theorem 3.1

The proof can be found in (Hardt et al., 2016). We provide the proof in Appendix for reference. Denote by $S = (z_1, \ldots, z_n)$ and $S' = (z_1', \ldots, z_n')$ two independent random samples and let $S^{(i)} = (z_1, \ldots, z_{i-1}, z_i', z_{i+1}, \ldots, z_n)$ be the sample that is identical to $S$ except in the $i$'th example where we replace $z_i$ with $z_i'$. With this notation, we get that

$$
\mathbb{E}_S \mathbb{E}_A \left[ R_S[A(S)] \right] = \mathbb{E}_S \mathbb{E}_A \left[ \frac{1}{n} \sum_{i=1}^n h(A(S); z_i) \right]
$$
$$
= \mathbb{E}_S \mathbb{E}_{S'} \mathbb{E}_A \left[ \frac{1}{n} \sum_{i=1}^n h(A(S^{(i)}); z_i') \right]
$$
$$
= \mathbb{E}_S \mathbb{E}_{S'} \mathbb{E}_A \left[ \frac{1}{n} \sum_{i=1}^n h(A(S); z_i') \right] + \delta
$$
$$
= \mathbb{E}_S \mathbb{E}_A \left[ R_{\mathcal{D}}[A(S)] \right] + \delta,
$$

where we can express $\delta$ as

$$
\delta = \mathbb{E}_S \mathbb{E}_{S'} \mathbb{E}_A \left[ \frac{1}{n} \sum_{i=1}^n h(A(S^{(i)}); z_i') - \frac{1}{n} \sum_{i=1}^n h(A(S); z_i') \right].
$$

Furthermore, taking the supremum over any two data sets $S, S'$ differing in only one sample, we can bound the difference as

$$
|\delta| \leq \sup_{S, S', z} \mathbb{E}_A \left[ h(A(S); z) - h(A(S'); z) \right] \leq \epsilon,
$$

by our assumption on the uniform stability of $A$. The claim follows.  □

## A.2  Proof of Lemma 4.1

Let the adversarial examples for parameter $\theta_1$ and $\theta_2$ be

$$
z_1 \in \arg \max_{\|z - z'\|_p \leq \epsilon} g(\theta_1, z')
$$
$$
z_2 \in \arg \max_{\|z - z'\|_p \leq \epsilon} g(\theta_2, z'),
$$

then we have

$$
\begin{aligned}
&\|h(\theta_1, z) - h(\theta_2, z)\| \\
=&|g(\theta_1, z_1) - g(\theta_2, z_2)| \\
\leq& \max\{|g(\theta_1, z_1) - g(\theta_2, z_1)|, |g(\theta_1, z_2) - g(\theta_2, z_2)|\} \\
\leq& L\|\theta_1 - \theta_2\|,
\end{aligned}
$$

where the first inequality is based on the fact that $g(\theta_1, z_1) \geq g(\theta_1, z_2)$ and $g(\theta_2, z_2) \geq g(\theta_2, z_1)$, the second inequality is based on Assumption 4.1. This proves Lemma 4.1.1.

For all subgradient $d(\theta, z) \in \partial_\theta h(\theta, z)$, we have

$$
\begin{aligned}
&\|d(\theta_1, z) - d(\theta_2, z)\| \\
=&\|\nabla_\theta g(\theta_1, z_1) - \nabla_\theta g(\theta_2, z_2)\| \\
\leq&\|\nabla_\theta g(\theta_1, z_1) - \nabla_\theta g(\theta_2, z_1)\| + \|\nabla_\theta g(\theta_2, z_1) - \nabla_\theta g(\theta_2, z_2)\| \\
\leq& L_\theta \|\theta_1 - \theta_2\| + L_z \|z_1 - z_2\|_p \\
\leq& L_\theta \|\theta_1 - \theta_2\| + L_z [\|z_1 - z\|_p + \|z - z_2\|_p] \\
\leq& L_\theta \|\theta_1 - \theta_2\| + 2 L_z \epsilon
\end{aligned}
$$

where the first and the third inequality is due to triangle inequality, the second inequality is based on Assumption 4.1. This proves the second inequality (non-gradient Lipschitz) in Lemma 4.1.  □

## A.3 Proof of Lemma 4.2

Proof of Lemma 4.2.1 ($\eta$-approximate descent Lemma).
Let $\tilde{\theta}$ be a point in the line segment of $\theta_1$ and $\theta_2$, $\tilde{\theta}(u) = \theta_2 + u(\theta_1 - \theta_2)$, then

$$h(\theta_1) - h(\theta_2)$$

$$= \int_0^1 \langle \theta_1 - \theta_2, \nabla_\theta h(\tilde{\theta}(u)) \rangle du$$

$$= \int_0^1 \langle \theta_1 - \theta_2, \nabla_\theta h(\theta_2) + \nabla_\theta h(\tilde{\theta}(u)) - \nabla_\theta h(\theta_2) \rangle du$$

$$= \langle \nabla_\theta h(\theta_2), \theta_1 - \theta_2 \rangle + \int_0^1 \langle \theta_1 - \theta_2, \nabla_\theta h(\tilde{\theta}(u)) - \nabla_\theta h(\theta_2) \rangle du$$

$$\leq \langle \nabla_\theta h(\theta_2), \theta_1 - \theta_2 \rangle + \int_0^1 \|\theta_1 - \theta_2\| \|\nabla_\theta h(\tilde{\theta}(u)) - \nabla_\theta h(\theta_2)\| du$$

$$\leq \langle \nabla_\theta h(\theta_2), \theta_1 - \theta_2 \rangle + \int_0^1 \|\theta_1 - \theta_2\| [\beta \|\tilde{\theta}(u) - \theta_2\| + \eta] du$$

$$= \langle \nabla_\theta h(\theta_2), \theta_1 - \theta_2 \rangle + \int_0^1 \|\theta_1 - \theta_2\| [\beta u \|\theta_1 - \theta_2\| + \eta] du$$

$$= \langle \nabla_\theta h(\theta_2), \theta_1 - \theta_2 \rangle + \beta \|\theta_1 - \theta_2\|^2 \int_0^1 u \, du + \eta \|\theta_1 - \theta_2\|]$$

$$= \langle \nabla_\theta h(\theta_2), \theta_1 - \theta_2 \rangle + \frac{\beta}{2} \|\theta_1 - \theta_2\|^2 + \eta \|\theta_1 - \theta_2\|.$$

$\square$

Proof of Lemma 4.2.2 ($\eta$-approximate co-coercive).
By Lemma 4.2.1 ($\eta$-approximate descent Lemma), we have

$$h(\theta_1) \leq h(\theta_2) + \langle \nabla_\theta h(\theta_2), \theta_1 - \theta_2 \rangle + \frac{\beta}{2} \|\theta_1 - \theta_2\|^2 + \eta \|\theta_1 - \theta_2\|.$$

Let $\theta^*$ be a minimizer of $h$, then

$$h(\theta^*) = \inf_{\theta_1} h(\theta_1) \leq \inf_{\theta_1} \left( h(\theta_2) + \langle \nabla_\theta h(\theta_2), \theta_1 - \theta_2 \rangle + \frac{\beta}{2} \|\theta_1 - \theta_2\|^2 + \eta \|\theta_1 - \theta_2\| \right)$$

$$= \inf_{\|v\|=1} \inf_{t \geq 0} \left( h(\theta_2) + t \nabla_\theta h(\theta_2)^T v + \frac{\beta t^2}{2} + \eta t \right),$$

where $t = \|\theta_1 - \theta_2\|$ and $v = (\theta_1 - \theta_2)/\|\theta_1 - \theta_2\|$. Then

$$\inf_{\|v\|=1} \inf_{t \geq 0} \left( h(\theta_2) + t \nabla_\theta h(\theta_2)^T v + \frac{\beta t^2}{2} + \eta t \right)$$

$$= \inf_{t \geq 0} \left( h(\theta_2) - t \|\nabla_\theta h(\theta_2)\| + \frac{\beta t^2}{2} + \eta t \right)$$

$$= h(\theta_2) + \inf_{t \geq 0} \left( - t(\|\nabla_\theta h(\theta_2)\| - \eta) + \frac{\beta t^2}{2} \right).$$

If $\|\nabla_\theta h(\theta_2)\| - \eta \leq 0$, the quaduatic function is optimized when $t = 0$. Then

$$h(\theta_2) + \inf_{t \geq 0} \left( - t(\|\nabla_\theta h(\theta_2)\| - \eta) + \frac{\beta t^2}{2} \right)$$

$$= h(\theta_2).$$

If $\|\nabla_\theta h(\theta_2)\| - \eta \geq 0$, then

$$h(\theta_2) + \inf_{t \geq 0} \left( - t(\|\nabla_\theta h(\theta_2)\| - \eta) + \frac{\beta t^2}{2} \right)$$

$$= h(\theta_2) - \frac{1}{2\beta} [\|\nabla_\theta h(\theta_2)\| - \eta]^2.$$

Therefore, we obtain that

$$h(\theta^*) - h(\theta) \leq -\frac{1}{2\beta}\Big[[\|\nabla h(\theta)\| - \eta]_+\Big]^2. \tag{A.1}$$

Define

$$h_1(w) = h(w) - \nabla h(\theta_1)^T w$$

and

$$h_2(w) = h(w) - \nabla h(\theta_2)^T w.$$

Firstly, it is easy to see that $h_1(w)$ and $h_2(w)$ are both $\eta$-approximate $\beta$-gradient Lipschitz, which satisfies inequatily in Eq. (A.1). Secondly, $w = \theta_1$ minimizes $h_1(w)$. Then

$$\begin{aligned}
&h(\theta_2) - h(\theta_1) - \nabla h(\theta_1)^T(\theta_2 - \theta_1) \\
=&h_1(\theta_2) - h_1(\theta_1) \\
\geq&\frac{1}{2\beta}\Big[[\|\nabla h_1(\theta_2)\| - \eta]_+\Big]^2 \\
=&\frac{1}{2\beta}\Big[[\|\nabla h(\theta_1) - \nabla h(\theta_2)\| - \eta]_+\Big]^2.
\end{aligned} \tag{A.2}$$

Similarly, we have

$$h(\theta_1) - h(\theta_2) - \nabla h(\theta_2)^T(\theta_1 - \theta_2) \geq \frac{1}{2\beta}\Big[[\|\nabla h(\theta_1) - \nabla h(\theta_2)\| - \eta]_+\Big]^2. \tag{A.3}$$

Take the summation of Eq. (A.2) and Eq. (A.3), we have

$$\langle \nabla h(\theta_1) - \nabla h(\theta_2), \theta_1 - \theta_2 \rangle \geq \frac{1}{\beta}\Big[[\|\nabla h(\theta_1) - \nabla h(\theta_2)\| - \eta]_+\Big]^2.$$

$\square$

## A.4  Proof of Lemma 4.3

Proof of Lemma 4.3.1 ($\alpha\eta$-approximately $(1 + \alpha\beta)$-expansive).

$$\begin{aligned}
&\|G_{\alpha,z}(\theta_1) - G_{\alpha,z}(\theta_2)\| \\
=&\|\theta_1 - \theta_2 - \alpha(\nabla h(\theta_1) - \nabla h(\theta_2))\| \\
\leq&\|\theta_1 - \theta_2\| + \|\alpha(\nabla h(\theta_1) - \nabla h(\theta_2))\| \\
\leq&\|\theta_1 - \theta_2\| + \alpha(\beta\|\theta_1 - \theta_2\| + \eta) \\
\leq&(1 + \alpha\beta)\|\theta_1 - \theta_2\| + \alpha\eta.
\end{aligned}$$

$\square$

Proof of Lemma 4.3.2 ($\alpha\eta$-approximately non-expansive.) Let t = $\|\nabla h(\theta_1) - \nabla h(\theta_2)\|$. If $t \leq \eta$, we have

$$\begin{aligned}
&\|G_{\alpha,z}(\theta_1) - G_{\alpha,z}(\theta_2)\| \\
=&\|\theta_1 - \theta_2 - \alpha(\nabla h(\theta_1) - \nabla h(\theta_2))\| \\
\leq&\|\theta_1 - \theta_2\| + \alpha\|\nabla h(\theta_1) - \nabla h(\theta_2)\| \\
\leq&\|\theta_1 - \theta_2\| + \alpha\eta.
\end{aligned}$$

If $t \geq \eta$, we have

$$
\begin{aligned}
&\|G_{\alpha,z}(\theta_1) - G_{\alpha,z}(\theta_2)\|^2 \\
=&\|\theta_1 - \theta_2 - \alpha(\nabla h(\theta_1) - \nabla h(\theta_2))\|^2 \\
=&\|\theta_1 - \theta_2\|^2 - 2\alpha(\nabla h(\theta_1) - \nabla h(\theta_2)^T(\theta_1 - \theta_2) + \alpha^2 t^2 \\
\leq&\|\theta_1 - \theta_2\|^2 - \frac{2\alpha}{\beta}(t-\eta)^2 + \alpha^2 t^2 \\
=&\|\theta_1 - \theta_2\|^2 - \frac{2\alpha}{\beta}(t-\eta)^2 - \frac{2\alpha\eta}{\beta}(t-\eta) + \alpha^2 t^2 + \frac{2\alpha\eta}{\beta}(t-\eta) \\
=&\|\theta_1 - \theta_2\|^2 - \frac{2\alpha t}{\beta}(t-\eta) + \alpha^2 t^2 + \frac{2\alpha\eta}{\beta}(t-\eta).
\end{aligned}
$$

Let $\alpha \leq 1/\beta$, then

$$
\begin{aligned}
&\|\theta_1 - \theta_2\|^2 - \frac{2\alpha t}{\beta}(t-\eta) + \alpha^2 t^2 + \frac{2\alpha\eta}{\beta}(t-\eta) \\
\leq&\|\theta_1 - \theta_2\|^2 - 2\alpha^2 t(t-\eta) + \alpha^2 t^2 + \frac{2\alpha\eta}{\beta}(t-\eta) \\
\leq&\|\theta_1 - \theta_2\|^2 - \alpha^2(t+\eta)(t-\eta) + \alpha^2 t^2 + \frac{2\alpha\eta}{\beta}(t-\eta) \\
\leq&\|\theta_1 - \theta_2\|^2 + \alpha^2\eta^2 + \frac{2\alpha\eta}{\beta}(t-\eta).
\end{aligned}
$$

By the definition of $\eta$-approximate smoothness,

$$
\frac{1}{\beta}(t-\eta) \leq \|\theta_1 - \theta_2\|.
$$

Then

$$
\begin{aligned}
&\|\theta_1 - \theta_2\|^2 + \alpha^2\eta^2 + \frac{2\alpha\eta}{\beta}(t-\eta) \\
\leq&\|\theta_1 - \theta_2\|^2 + \alpha^2\eta^2 + 2\alpha\eta\|\theta_1 - \theta_2\| \\
=&(\|\theta_1 - \theta_2\| + \alpha\eta)^2.
\end{aligned}
$$

Therefore, we obtain that

$$
\|G_{\alpha,z}(\theta_1) - G_{\alpha,z}(\theta_2)\| \leq \|\theta_1 - \theta_2\| + \alpha\eta.
$$

$\square$

Proof of Lemma 4.3.3 ($\alpha\eta$-approximately $(1 - \alpha\gamma)$-contraction.).
Firstly, if $h(\theta)$ is a $\gamma$-strongly convex, $\eta$-approximately $\beta$-gradient Lipschitz function, $\phi(\theta) = h(\theta) - \frac{\gamma}{2}\|\theta\|^2$ is a convex, $\eta$-approximate $(\beta-\gamma)$-gradient Lipschitz function. The proof of convexity follows the definition. To see the second claim, since

$$
\begin{aligned}
&\phi(\theta_1) - \phi(\theta_2) \\
=&h(\theta_1) - h(\theta_2) - (\frac{\gamma}{2}\|\theta_1\|^2 - \frac{\gamma}{2}\|\theta_2\|^2) \\
\leq&\langle\nabla_\theta h(\theta_2), \theta_1 - \theta_2\rangle + \frac{\beta}{2}\|\theta_1 - \theta_2\|^2 + \eta\|\theta_1 - \theta_2\| - (\frac{\gamma}{2}\|\theta_1\|^2 - \frac{\gamma}{2}\|\theta_2\|^2) \\
\leq&\langle\nabla_\theta \phi(\theta_2), \theta_1 - \theta_2\rangle + \frac{\beta}{2}\|\theta_1 - \theta_2\|^2 + \eta\|\theta_1 - \theta_2\| - (\frac{\gamma}{2}\|\theta_1\|^2 - \frac{\gamma}{2}\|\theta_2\|^2) + \gamma\theta_2^T(\theta_1 - \theta_2) \\
\leq&\langle\nabla_\theta \phi(\theta_2), \theta_1 - \theta_2\rangle + \frac{\beta}{2}\|\theta_1 - \theta_2\|^2 + \eta\|\theta_1 - \theta_2\| - \frac{\gamma}{2}\|\theta_1 - \theta_2\|^2 \\
\leq&\langle\nabla_\theta \phi(\theta_2), \theta_1 - \theta_2\rangle + \frac{\beta - \gamma}{2}\|\theta_1 - \theta_2\|^2 + \eta\|\theta_1 - \theta_2\|.
\end{aligned}
$$

Therefore, $\phi(\theta)$ satisfies the $\eta$-approximate $(\beta - \gamma)$-descent Lemma. Let $t = \|\nabla\phi(\theta_1) - \nabla\phi(\theta_2)\|$. If $t \leq \eta$, we have

$$
\begin{aligned}
&\|G_{\alpha,z}(\theta_1) - G_{\alpha,z}(\theta_2)\| \\
=&\|\theta_1 - \theta_2 - \alpha(\nabla h(\theta_1) - \nabla h(\theta_2))\| \\
=&\|\theta_1 - \theta_2 - \alpha(\nabla\phi(\theta_1) + \gamma\theta_1 - \nabla\phi(\theta_2) - \gamma\theta_2)\| \\
\leq&\|(1 - \alpha\gamma)(\theta_1 - \theta_2)\| + \alpha\|\nabla\phi(\theta_1) - \nabla\phi(\theta_2)\| \\
\leq&(1 - \alpha\gamma)\|\theta_1 - \theta_2\| + \alpha\eta.
\end{aligned}
$$

If $t \geq \eta$, we have

$$
\begin{aligned}
&\|G_{\alpha,z}(\theta_1) - G_{\alpha,z}(\theta_2)\|^2 \\
=&\|\theta_1 - \theta_2 - \alpha(\nabla h(\theta_1) - \nabla h(\theta_2))\|^2 \\
=&\|\theta_1 - \theta_2 - \alpha(\nabla\phi(\theta_1) + \gamma\theta_1 - \nabla\phi(\theta_2) - \gamma\theta_2)\|^2 \\
\leq&(1 - \alpha\gamma)^2\|\theta_1 - \theta_2\|^2 - 2\alpha(1 - \alpha\gamma)(\nabla\phi(\theta_1) - \nabla\phi(\theta_2)^T(\theta_1 - \theta_2) + \alpha^2 t^2 \\
\leq&(1 - \alpha\gamma)^2\|\theta_1 - \theta_2\|^2 - \frac{2\alpha(1 - \alpha\gamma)}{\beta - \gamma}(t - \eta)^2 + \alpha^2 t^2 \\
\leq&(1 - \alpha\gamma)^2\|\theta_1 - \theta_2\|^2 - \frac{2\alpha(1 - \alpha\gamma)}{\beta - \gamma}(t - \eta)^2 - \frac{2\alpha(1 - \alpha\gamma)\eta}{\beta - \gamma}(t - \eta) + \alpha^2 t^2 + \frac{2\alpha(1 - \alpha\gamma)\eta}{\beta - \gamma}(t - \eta).
\end{aligned}
$$

Since $\alpha \leq 1/\beta$, we have $(1 - \alpha\gamma)/(\beta - \gamma) \geq \alpha$, then

$$
\begin{aligned}
&(1 - \alpha\gamma)^2\|\theta_1 - \theta_2\|^2 - \frac{2\alpha(1 - \alpha\gamma)}{\beta - \gamma}(t - \eta)^2 - \frac{2\alpha(1 - \alpha\gamma)\eta}{\beta - \gamma}(t - \eta) + \alpha^2 t^2 + \frac{2\alpha(1 - \alpha\gamma)\eta}{\beta - \gamma}(t - \eta) \\
\leq&(1 - \alpha\gamma)^2\|\theta_1 - \theta_2\|^2 - \alpha^2 t(t - \eta) + \alpha^2 t^2 + \frac{2\alpha(1 - \alpha\gamma)\eta}{\beta - \gamma}(t - \eta) \\
\leq&(1 - \alpha\gamma)^2\|\theta_1 - \theta_2\|^2 + \alpha^2\eta^2 + 2\alpha(1 - \alpha\gamma)\eta\|\theta_1 - \theta_2\| \\
\leq&\left((1 - \alpha\gamma)\|\theta_1 - \theta_2\| + \alpha\eta\right)^2.
\end{aligned}
$$

Therefore, we obtain that

$$
\|G_{\alpha,z}(\theta_1) - G_{\alpha,z}(\theta_2)\| \leq (1 - \alpha\gamma)\|\theta_1 - \theta_2\| + \alpha\eta.
$$

$\square$

## A.5 Proof of Theorem 5.1

The proof follows the standard techniques for uniform stability. We need to replace the non-expansive property used in standard analysis by the approximately non-expansive property. Let $S$ and $S'$ be two samples of size $n$ differing in only a single example. Consider two trajectories $\theta_1^1, \ldots, \theta_1^T$ and $\theta_2^1, \ldots, \theta_2^T$ induced by running SGD on sample $S$ and $S'$, respectively. Let $\delta_t = \|\theta_1^t - \theta_2^t\|$.

Fixing an example $z \in Z$ and apply the Lipschitz condition on $h(\cdot; z)$, we have

$$
\mathbb{E}\left|h(\theta_1^T; z) - h(\theta_2^T; z)\right| \leq L\,\mathbb{E}\left[\delta_T\right]. \tag{A.4}
$$

Observe that at step $t$, with probability $1 - 1/n$, the example selected by SGD is the same in both $S$ and $S'$. With probability $1/n$ the selected example is different. Therefore, by the $\alpha\eta$-approximate non-expansive property, we have

$$
\mathbb{E}\left[\delta_{t+1}\right] \leq \left(1 - \frac{1}{n}\right)\left(\mathbb{E}\left[\delta_t\right] + \alpha_t\eta\right) + \frac{1}{n}\mathbb{E}\left[\delta_t\right] + \frac{2\alpha_t L}{n} \leq \mathbb{E}\left[\delta_t\right] + \left(\eta + \frac{2L}{n}\right)\alpha_t. \tag{A.5}
$$

Unraveling the recursion gives

$$
\mathbb{E}\left[\delta_T\right] \leq \left(\eta + \frac{2L}{n}\right)\sum_{t=1}^{T}\alpha_t.
$$

Plugging this back into Eq. (A.4), we obtain

$$\mathcal{E}_{gen} \leq L\left(\eta + \frac{2L}{n}\right)\sum_{t=1}^{T}\alpha_t.$$

Since this bounds holds for all $S, S'$ and $z$, we obtain the desired bound on the uniform stability. $\quad\square$

## A.6 Proof of Theorem 5.2

Proof: The construction of function $h$ is adopted from the construction in the work of (Bassily et al., 2020).

Let $T \leq d$, and $v, K \geq 0$. Considering $\mathcal{Z} = \{0, 1\}$, and the objective function

$$h(\theta, z) = \begin{cases} \eta \max\{0, x_1 - v, \cdots, x_T - v\} & \text{if } z = 0 \\ \langle r, x \rangle / K & \text{if } z = 1, \end{cases} \tag{A.6}$$

where $r = (-1, \cdots, -1, 0, \cdots, 0)$ (i.e., equals to -1 for the first $T$ components). Function $h$ is $\eta$-approximately smooth since the first case is a piece-wise linear function. For the dataset $S$ and $S'$ differ in at most one sample, the empirical objective functions are

$$R_S(\theta) = \frac{1}{nK}\langle r, x \rangle + \frac{n-1}{n}\eta \max\{0, x_1 - v, \cdots, x_T - v\},$$

and

$$R_{S'}(\theta) = \eta \max\{0, x_1 - v, \cdots, x_T - v\}.$$

Let $\theta_1^T$ and $\theta_2^T$ be the trajectories running algorithm on dataset $S$ and $S'$, initialized on $\theta_1^0 = \theta_2^0 = 0$. Clearly, $\theta_2^t = 0$ for all $t$. It is easy to obtain $\theta_1^t = -\frac{t\alpha r}{nk} - \alpha\eta\frac{n-1}{n}\sum_{s=1}^{t-1}e_s$ recursively. By the orthogonality of the subgradients, we have

$$\delta(S, S') = \|\theta_1^T - \theta_2^T\| = \|\theta_1^T\| \geq \Omega\left(\alpha\eta\left\|\sum_{s=1}^{T}e_s\right\|\right) = \Omega(\alpha\eta\sqrt{T}). \tag{A.7}$$

On the other hand, the work of (Hardt et al., 2016) provided a lower bound for general non-smooth function

$$\delta(S, S') \geq \Omega\left(\frac{L\alpha T}{n}\right). \tag{A.8}$$

Combining Eq. (A.7) and Eq. (A.8), we have

$$\delta(S, S') \geq \Omega\left(\alpha\eta\sqrt{T} + \frac{L\alpha T}{n}\right). \tag{A.9}$$

$\square$

## A.7 Proof of Theorem 5.3

We consider a general form of Theorem 5.3.

**Theorem A.1** (Non-convex). *Assume that $h(\theta, z)$ is $L$-Lipschitz, and $\eta$-approximately $\beta$-gradient Lipschitz in $\theta$ for all given $z \in \mathcal{Z}$. Assume in addition that $0 \leq g(\theta, z) \leq B$ for all $\theta$ and $z$. Suppose that we run SGD on the adversarial surrogate loss with step sizes $\alpha_t \leq c/t$ for $T$ steps, where $c > 0$. Then, for all $t_0 \in \{1, 2, \cdots, n\}$, adversarial training satisfies uniform stability with*

$$\mathcal{E}_{gen} = \mathbb{E}[R_\mathcal{D}(\theta^T) - R_S(\theta^T)] \leq \frac{Bt_0}{n-1} + \frac{1}{\beta(n-1)}\left(2L^2 + L\eta n\right)\left(\frac{T}{t_0}\right)^{\beta c}. \tag{A.10}$$

*Let $q = \beta c$. For small $T$ (s.t. $t_0 \leq n$ when we set the first term equals to the second term). we select $t_0$ to optimize the right hand side, then*

$$\mathcal{E}_{gen} \leq \frac{2}{n-1}B^{\frac{q}{q+1}}\left(\frac{2L^2 + L\eta n}{\beta}\right)^{\frac{1}{q+1}}T^{\frac{q}{q+1}}. \tag{A.11}$$

*For arbitrary $T$, the optimal $t_0 > n$. We simply let $t_0 = 1$, then*

$$\mathcal{E}_{gen} \leq \frac{BL_\theta + (2L^2 + L\eta n)T^q}{\beta(n-1)}. \tag{A.12}$$

Proof: Let $S$ and $S'$ be two samples of size $n$ differing in only a single example. Consider two trajectories $\theta_1^1, \ldots, \theta_1^T$ and $\theta_2^1, \ldots, \theta_2^T$ induced by running SGD on sample $S$ and $S'$, respectively. Let $\delta_t = \|\theta_1^t - \theta_2^t\|$. Let $t_0 \in \{0, 1, \ldots, n\}$, be the iteration that $\delta_{t_0} = 0$, but SGD picks two different samples form $S$ and $S'$ in iteration $t_0 + 1$, then

$$\mathcal{E}_{gen} \leq \frac{t_0}{n} B + L \,\mathbb{E}\left[\delta_T \mid \delta_{t_0} = 0\right] . \tag{A.13}$$

Let $\Delta_t = \mathbb{E}\left[\delta_t \mid \delta_{t_0} = 0\right]$. Observe that at step $t$, with probability $1 - 1/n$, the example selected by SGD is the same in both $S$ and $S'$. With probability $1/n$ the selected example is different. Therefore, by the $\alpha\eta$-approximate $(1 + \alpha\beta)$-expansive property, for every $t \geq t_0$,

$$\Delta_{t+1} \leq \left(1 - \frac{1}{n}\right)(1 + \alpha_t\beta)\Delta_t + \frac{1}{n}\Delta_t + \left(\eta + \frac{2L}{n}\right)\alpha_t$$

$$\leq \left(\frac{1}{n} + (1 - 1/n)(1 + c\beta/t)\right)\Delta_t + \left(\eta + \frac{2L}{n}\right)\frac{c}{t}$$

$$= \left(1 + (1 - 1/n)\frac{c\beta}{t}\right)\Delta_t + \left(\eta + \frac{2L}{n}\right)\frac{c}{t}$$

$$\leq \exp\left((1 - 1/n)\frac{c\beta}{t}\right)\Delta_t + \left(\eta + \frac{2L}{n}\right)\frac{c}{t} .$$

Here we used the fact that $1 + x \leq \exp(x)$ for all $x$.

Using the fact that $\Delta_{t_0} = 0$, we can unwind this recurrence relation from $T$ down to $t_0 + 1$. This gives

$$\Delta_T \leq \sum_{t=t_0+1}^{T} \left\{ \prod_{k=t+1}^{T} \exp\left((1 - \tfrac{1}{n})\frac{\beta c}{k}\right) \right\} \left(\eta + \frac{2L}{n}\right)\frac{c}{t}$$

$$= \sum_{t=t_0+1}^{T} \exp\left((1 - \tfrac{1}{n})\beta c \sum_{k=t+1}^{T} \tfrac{1}{k}\right) \left(\eta + \frac{2L}{n}\right)\frac{c}{t}$$

$$\leq \sum_{t=t_0+1}^{T} \exp\left((1 - \tfrac{1}{n})\beta c \log(\tfrac{T}{t})\right) \left(\eta + \frac{2L}{n}\right)\frac{c}{t}$$

$$= \left(\eta + \frac{2L}{n}\right)cT^{\beta c(1-1/n)} \sum_{t=t_0+1}^{T} t^{-\beta c(1-1/n)-1}$$

$$\leq \left(\eta + \frac{2L}{n}\right)\frac{1}{(1 - 1/n)\beta c}c\left(\frac{T}{t_0}\right)^{\beta c(1-1/n)}$$

$$\leq \frac{\eta n + 2L}{\beta(n-1)}\left(\frac{T}{t_0}\right)^{\beta c} ,$$

Plugging this bound into (A.13), we get

$$\mathcal{E}_{gen} \leq \frac{Bt_0}{n-1} + \frac{L\eta n + 2L^2}{\beta(n-1)}\left(\frac{T}{t_0}\right)^{\beta c} .$$

Let $q = \beta c$. For small $T$ (s.t. $t_0 \leq n$ when we set the first term equals to the second term). we select $t_0$ to optimize the right hand side, then

$$\mathcal{E}_{gen} \leq \frac{2}{n-1} B^{\frac{q}{q+1}} \left(\frac{2L^2 + L\eta n}{\beta}\right)^{\frac{1}{q+1}} T^{\frac{q}{q+1}} .$$

For arbitrary $T$, the optimal $t_0 > n$. We simply let $t_0 = 1$, then

$$\mathcal{E}_{gen} \leq \frac{BL_\theta + (2L^2 + L\eta n)T^q}{\beta(n-1)} .$$

Since the bound we just derived holds for all $S, S'$ and $z$, we immediately get the claimed upper bound on the uniform stability. Let $q = 1$, we obtain the result of Theorem 5.3. $\qquad \square$

## A.8 Proof of Theorem 5.4

The proof follows the idea in convex case. By the $\alpha\eta$-approximately $(1-\alpha\gamma)$-contraction, for every $t$,

$$\mathbb{E}\,\delta_{t+1} \leq \left(1 - \frac{1}{n}\right)(1-\alpha\gamma)\,\mathbb{E}\,\delta_t + \frac{1}{n}(1-\alpha\gamma)\,\mathbb{E}\,\delta_t + \left(\eta + \frac{2L}{n}\right)\alpha \qquad (A.14)$$

$$= (1-\alpha\gamma)\,\mathbb{E}\,\delta_t + \left(\eta + \frac{2L}{n}\right)\alpha\,.$$

Unraveling the recursion gives

$$\mathbb{E}\,\delta_T \leq \left(\eta + \frac{2L}{n}\right)\alpha \sum_{t=0}^{T}(1-\alpha\gamma)^t \leq \frac{\eta}{\gamma} + \frac{2L}{\gamma n}\,.$$

Plugging the above inequality into Eq. (A.4), we obtain

$$\mathcal{E}_{gen} \leq \frac{L\eta}{\gamma} + \frac{2L^2}{\gamma n}\,.$$

Since this bounds holds for all $S, S'$ and $z$, the Theorem follows. $\qquad\square$

## A.9 Proof of Theorem 6.2

Let $\bar{\theta} = \frac{1}{T}\sum_{t=1}^{T}\theta^t$ denote the average of the stochastic gradient iterates. Since

$$\theta^t = \sum_{k=1}^{t}\alpha_k \nabla h(\theta^k; z_k)\,,$$

we have

$$\bar{\theta} = \sum_{t=1}^{T}\alpha_t \frac{T-t+1}{T}\nabla h(\theta^k; z_k)$$

Using the $\alpha\eta$-approximate non-expansive, we have

$$\delta_t \leq (1-1/n)\delta_{t-1} + \frac{1}{n}\left(\delta_{t-1} + (\eta n + 2L)\alpha_t \frac{T-t+1}{T}\right)\,.$$

which implies

$$\delta_T \leq \left(\eta + \frac{2L}{n}\right)\sum_{t=1}^{T}\alpha_t \frac{T-t+1}{T} = \left(\frac{\eta}{2} + \frac{L}{n}\right)\sum_{t=1}^{T}\alpha_t\,.$$

Since $f$ is $L$-Lipschitz, we have

$$\mathcal{E}_{gen}(\bar{\theta}) \leq \left(\frac{L\eta}{2} + \frac{L^2}{n}\right)\sum_{t=1}^{T}\alpha_t. \qquad (A.15)$$

Here the expectation is taken over the algorithm and hence the claim follows by our definition of uniform stability. $\mathcal{E}_{opt}$ follows (Nemirovski et al., 2009). $\qquad\square$

# B Discussion on Non-convex and Strongly Convex Case

## B.1 Discussion on Non-convex Case

To discuss the generalization-optimization trade-off in the non-convex case. We first give the optimization error bound.

**Theorem B.1.** *Assume that $h$ is $\eta$-approximate $\beta$-gradient Lipschitz and given $0 < \tau < 1$. Without loss of generality, assume the stochastic gradient $\nabla\tilde{h}(\theta)$ be unbiased and have a bounded variance $\sigma^2$. Let the stochastic gradient descent (SGD) update be $\theta_{t+1} = \theta_t - \alpha\nabla\tilde{h}(\theta_t)$ with a constant step size $\alpha = 1/\sqrt{T}$ for number of iterations $T \geq (\beta/2(1-\tau))^2$. $\exists t \leq T$, s.t.*

$$\mathbb{E}\|\nabla h(\theta_t)\|^2 \leq \frac{\eta^2}{\tau^2} + \frac{2\eta\sigma}{\tau} + \mathcal{O}(\frac{1}{\sqrt{T}}). \qquad (B.1)$$

Proof:

Assume that the stochastic gradient $\nabla h(\theta)$ be unbiased and have a bounded variance $\sigma^2$.

$$\mathbb{E}[\nabla \tilde{h}(\theta)] = \nabla h(\theta),$$

$$\mathbb{E}\|\nabla \tilde{h}(\theta)\|^2 \le \|\nabla h(\theta)\|^2 + \sigma^2.$$

Notice that when $\theta$ is a random vector, the above expectation is condition on $\theta$. Let the stochastic gradient descent (SGD) update be $\theta_{t+1} = \theta_t - \alpha \tilde{h}(\theta_t)$ with a constant step size $\alpha = 1/\sqrt{T}$. By $\eta$-approximately Descent Lemma, we have

$$h(\theta_{t+1}) - h(\theta_t)$$

$$= -\alpha\langle\nabla h(\theta_t), \theta_{t+1} - \theta_t\rangle + \frac{\beta}{2}\|\theta_{t+1} - \theta_t\|^2 + \eta\|\theta_{t+1} - \theta_t\|$$

$$= -\alpha\langle\nabla h(\theta_t), \nabla\tilde{h}(\theta_t)\rangle + \frac{\beta\alpha^2}{2}\|\nabla\tilde{h}(\theta_t)\|^2 + \eta\alpha\|\nabla\tilde{h}(\theta_t)\|.$$

Given $\theta_t$, take the conditional expectation over the noised introduced by SGD, we have

$$\mathbb{E}[h(\theta_{t+1})] - h(\theta_t)$$

$$= -\alpha\langle\nabla h(\theta_t), \mathbb{E}[\nabla\tilde{h}(\theta_t)]\rangle + \frac{\beta\alpha^2}{2}\mathbb{E}\|\nabla\tilde{h}(\theta_t)\|^2 + \eta\alpha\mathbb{E}\|\nabla\tilde{h}(\theta_t)\|$$

$$\le -\alpha\|\nabla h(\theta_t)\|^2 + \frac{\beta\alpha^2}{2}\left[\|\nabla h(\theta_t)\|^2 + \sigma^2\right] + \eta\alpha\sqrt{\left[\mathbb{E}\|\nabla\tilde{h}(\theta_t)\|\right]^2}$$

$$\le -\alpha\|\nabla h(\theta_t)\|^2 + \frac{\beta\alpha^2}{2}\left[\|\nabla h(\theta_t)\|^2 + \sigma^2\right] + \eta\alpha\sqrt{\|\nabla h(\theta_t)\|^2 + \sigma^2}$$

$$\le -\alpha\|\nabla h(\theta_t)\|^2 + \frac{\beta\alpha^2}{2}\left[\|\nabla h(\theta_t)\|^2 + \sigma^2\right] + \eta\alpha\left[\|\nabla h(\theta_t)\| + \sigma\right]$$

$$= -\alpha\|\nabla h(\theta_t)\|^2 + \frac{\beta\alpha^2}{2}\|\nabla h(\theta_t)\|^2 + \eta\alpha\|\nabla_\theta h(\theta_t)\| + \frac{\beta\alpha^2\sigma^2}{2} + \eta\alpha\sigma$$

$$\le -\tau\alpha\|\nabla h(\theta_t)\|^2 + \eta\alpha\|\nabla h(\theta_t)\| + \frac{\beta\alpha^2\sigma^2}{2} + \eta\alpha\sigma,$$

where the first inequality is the assumption of SGD, the second inequality is the Jensen's inequality, the third one is the assumption of SGD, the fourth one is the Cauchy-Schwartz inequality, and the last one is because of the size of step size $\alpha$. Take the expectation over the trajectory $\theta_0, \theta_1, \cdots, \theta_T$, and take the average aver $t = 0, 1, \cdots, T$, we have

$$\frac{1}{T}\sum_{t=0}^{T}\left[\tau\alpha\mathbb{E}\|\nabla h(\theta_t)\|^2 - \eta\alpha\mathbb{E}\|\nabla h(\theta_t)\|\right]$$

$$\le \frac{1}{T}\sum_{t=0}^{T}\left[\mathbb{E}[h(\theta_t)] - \mathbb{E}h(\theta_{t+1})\right] + \frac{\beta\alpha^2\sigma^2}{2} + \eta\alpha\sigma$$

$$\le \frac{1}{T}\left[\mathbb{E}[h(\theta_0)] - h(\theta_*)\right] + \frac{\eta\alpha^2\sigma^2}{2} + \eta\alpha\sigma.$$

Let $\mathbb{E}[h(\theta_0)] - h(\theta_*) = D$ and divide $\alpha$ on both side. $\exists t \le T, s.t.$

$$\tau\mathbb{E}\|\nabla h(\theta_t)\|^2 - \eta\mathbb{E}\|\nabla h(\theta_t)\| \le \frac{D}{T\alpha} + \frac{\beta\alpha\sigma^2}{2} + \eta\sigma.$$

Since $\alpha = 1/\sqrt{T}$, we have

$$\tau\mathbb{E}\|\nabla h(\theta_t)\|^2 - \eta\mathbb{E}\|\nabla h(\theta_t)\| \le \frac{D}{\sqrt{T}} + \frac{\beta\sigma^2}{2\sqrt{T}} + \eta\sigma$$

$$\Leftrightarrow \mathbb{E}\|\nabla h(\theta_t)\|^2 - \frac{\eta}{\tau}\mathbb{E}\|\nabla h(\theta_t)\| + (\frac{\eta}{2\tau})^2 \le \frac{1}{\sqrt{T}}\left[\frac{D}{\tau} + \frac{\beta\sigma^2}{2\tau}\right] + \frac{\eta\sigma}{\tau} + (\frac{\eta}{2\tau})^2$$

$$\Leftrightarrow \left|\mathbb{E}\|\nabla h(\theta_t)\| - \frac{\eta}{2\tau}\right| \le \sqrt{\frac{1}{\sqrt{T}}\left[\frac{D}{\tau} + \frac{\beta\sigma^2}{2\tau}\right] + \frac{\eta\sigma}{\tau} + (\frac{\eta}{2\tau})^2}.$$

Then we remove the absolute value, and obtain

$$\mathbb{E}\|\nabla h(\theta_t)\| - \frac{\eta}{2\tau} \leq \sqrt{\frac{1}{\sqrt{T}}\left[\frac{D}{\tau} + \frac{\beta\sigma^2}{2\tau}\right] + \frac{\eta\sigma}{\tau} + (\frac{\eta}{2\tau})^2}$$

$$\Leftrightarrow \mathbb{E}\|\nabla h(\theta_t)\| \leq \frac{\eta}{2\tau} + \sqrt{\frac{1}{\sqrt{T}}\left[\frac{D}{\tau} + \frac{\beta\sigma^2}{2\tau}\right] + \frac{\eta\sigma}{\tau} + (\frac{\eta}{2\tau})^2}$$

$$\Leftrightarrow \mathbb{E}\|\nabla h(\theta_t)\|^2 \leq \left[\frac{\eta}{2\tau} + \sqrt{\frac{1}{\sqrt{T}}\left[\frac{D}{\tau} + \frac{\beta\sigma^2}{2\tau}\right] + \frac{\beta\sigma}{\tau} + (\frac{\eta}{2\tau})^2}\right]^2.$$

By Cauchy-Schwartz inequality, we have

$$\mathbb{E}\|\nabla h(\theta_t)\|^2 \leq \left[\frac{\eta}{2\tau} + \sqrt{\frac{1}{\sqrt{T}}\left[\frac{D}{\tau} + \frac{\beta\sigma^2}{2\tau}\right] + \frac{\eta\sigma}{\tau} + (\frac{\eta}{2\tau})^2}\right]^2$$

$$\leq 2\left[(\frac{\eta}{2\tau})^2 + \frac{1}{\sqrt{T}}\left[\frac{D}{\tau} + \frac{\beta\sigma^2}{2\tau}\right] + \frac{\beta\sigma}{\tau} + (\frac{\eta}{2\tau})^2\right]$$

$$= \frac{\eta^2}{\tau^2} + \frac{2\eta\sigma}{\tau} + \frac{2}{\sqrt{T}}\left[\frac{D}{\tau} + \frac{\beta\sigma^2}{2\tau}\right]$$

$$= \frac{\eta^2}{\tau^2} + \frac{2\eta\sigma}{\tau} + \mathcal{O}(\frac{1}{\sqrt{T}}).$$

$\square$

In words, running SGD on an approximately smooth non-convex function, the algorithm cannot converge to a stationary point but with an additional constant term. Notice that this is an error bound for gradient norm. If we need an error bound for the optimality gap, we need an additional PL condition. Combining the optimization error and generalization error, we have

$$\mathcal{E}_{opt} + \mathcal{E}_{gen} \leq \mathcal{O}\left(\eta T + \frac{T}{n} + \frac{1}{\sqrt{T}}\right) + \text{constant},$$

where the first term is an additional term for adversarial training, which induces robust overfitting. Therefore, we can see that the analysis of the convex and non-convex cases do not have a major difference. To simplify the argument, we only discuss the convex case in the main paper.

## B.2 Discussion on Strongly Convex Case

By (Nemirovski et al., 2009),

$$\mathcal{E}_{opt} \leq \frac{LD^2}{T} = \mathcal{O}(\frac{1}{T})$$

in the strongly convex case. Whether the function is smooth does not affect the convergence rate. Therefore,

$$\mathcal{E}_{opt} + \mathcal{E}_{gen} \leq \mathcal{O}\left(\eta + \frac{1}{n} + \frac{1}{T}\right).$$

This result shows that robust overfitting will disappear if the loss function is strongly convex. But the performance of adversarial training is still worse than the performance of standard training in $\mathcal{O}(\eta)$ in this strong assumption.

## B.3 Discussion on Strongly Concave Assumption on the Inner Problem

In this subsection, we discuss the case that $g(\theta, z)$ is $\mu$-strongly concave in $z$.

**Assumption B.1.** *The function g satisfies the following Lipschitzian smoothness conditions:*

$$\|g(\theta_1, z) - g(\theta_2, z)\| \leq L\|\theta_1 - \theta_2\|,$$
$$\|\nabla_\theta g(\theta_1, z) - \nabla_\theta g(\theta_2, z)\| \leq L_\theta\|\theta_1 - \theta_2\|,$$
$$\|\nabla_\theta g(\theta, z_1) - \nabla_\theta g(\theta, z_2)\| \leq L_z\|z_1 - z_2\|,$$
$$\|\nabla_z g(\theta_1, z) - \nabla_\theta g(\theta_2, z)\| \leq L_{z\theta}\|\theta_1 - \theta_2\|.$$

Assumption B.1 assumes that the loss function is smooth (in zeroth-order and first-order), which are also used in the stability literature (Farnia and Ozdaglar, 2021; Xing et al., 2021a), as well as the convergence analysis literature (Wang et al., 2019; Liu et al., 2020). Comparing with Assumption 4.1, Assumption B.1 requires one more gradient Lipschitz $\|\nabla_z g(\theta_1, z) - \nabla_\theta g(\theta_2, z)\| \le L_{z\theta}\|\theta_1 - \theta_2\|$.

**Lemma B.1.** *Under Assumption B.1, assume in addition that $g(\theta, z)$ is $\mu$-strongly concave in $z$. $\forall \theta_1, \theta_2$ and $\forall z \in \mathcal{Z}$, the following properties hold.*

1. *(Lipschitz function.)* $\|h(\theta_1, z) - h(\theta_2, z)\| \le L\|\theta_1 - \theta_2\|$.

2. *(gradient Lipschitz.)* $\|\nabla_\theta h(\theta_1, z) - \nabla_\theta h(\theta_2, z)\| \le \beta_2 \|\theta_1 - \theta_2\|$, *where*

$$\beta_2 = \frac{L_z L_{z\theta}}{\mu} + L_\theta.$$

The proof can be found in (Sinha et al., 2017; Wang et al., 2019). Therefore, the adversarial surrogate loss is $\beta_2$-gradient Lipschitz. The stability generalization bounds follows (Hardt et al., 2016) by replacing $\beta$ by $\beta_2$ (for the choice of step size $\alpha$).

## C   Additional Experiments

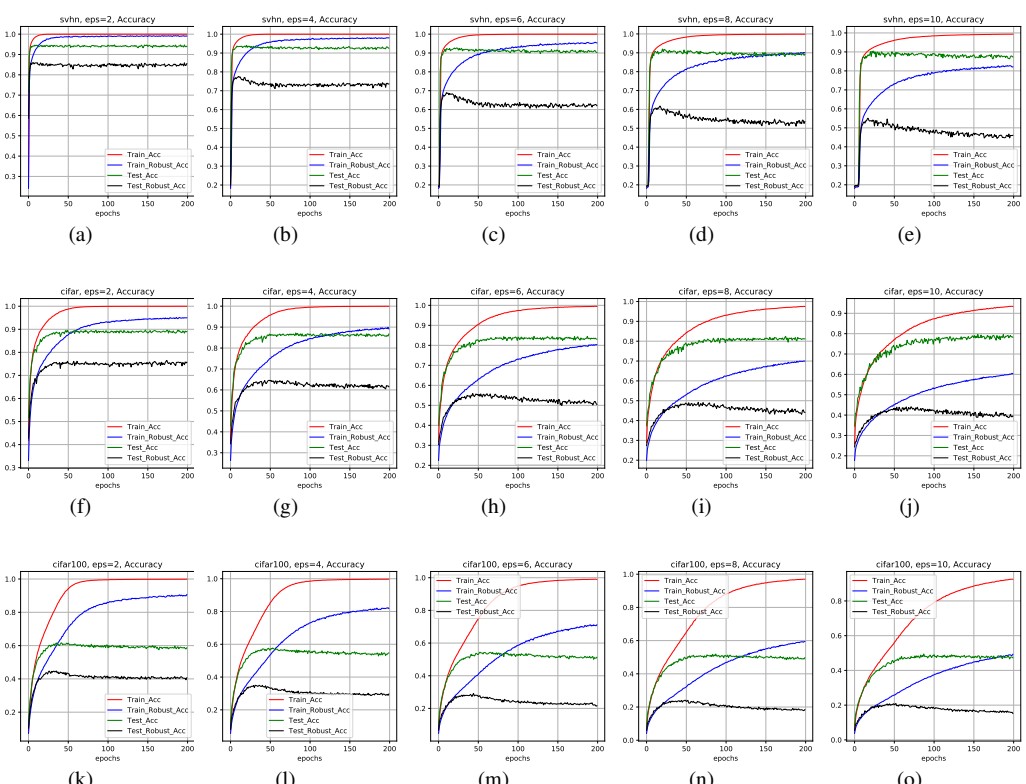

Figure 5: Accuracy of adversarial training with fixed learning rate = 0.01. The first row is the experiments on SVHN. The second row is the experiments on CIFAR-10. The last row is the experiments on CIFAR-100. The first column to the last column are the experiments of $\epsilon$ equal to 2, 4, 6, 8, and 10, respectively.

In this section, we provide additional experiments on SVHN, CIFAR-10, and CIFAR-100. In Fig. 5, we show the experiments of adversarial training using a fixed learning rate. In Fig. 6, we show the experiments of adversarial training using a standard piece-wise linear learning rate. In Fig. 7, we show the experiments of adversarial training using cyclic learning rate.

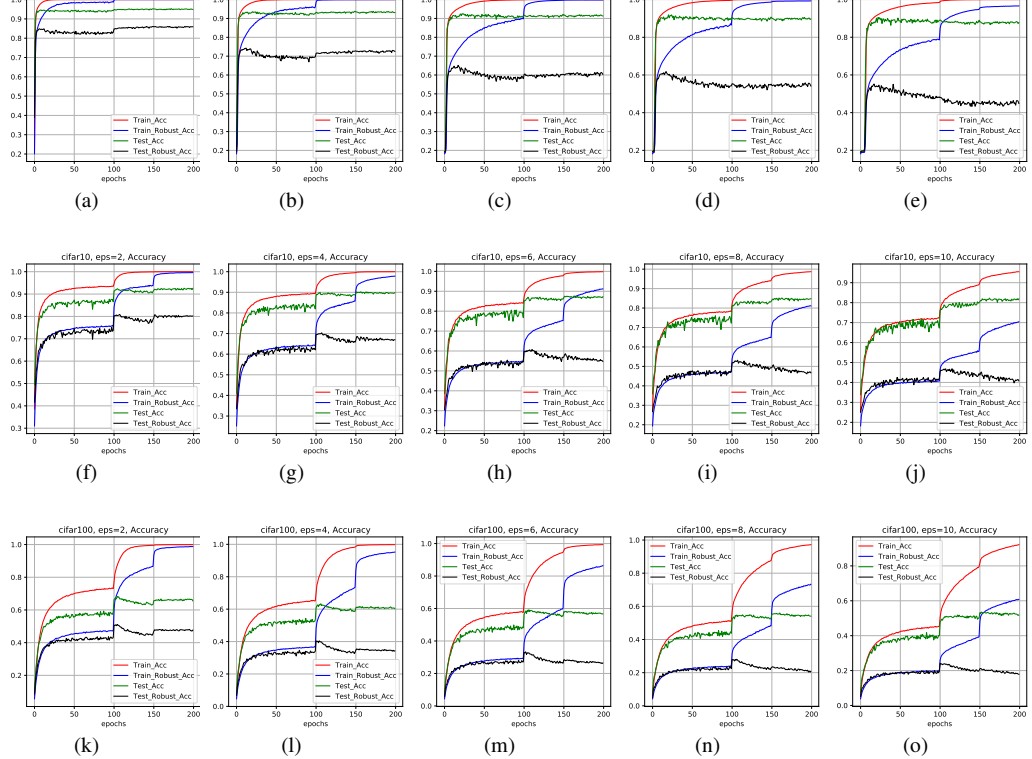

Figure 6: Accuracy of adversarial training with piece-wise linear learning rate. The first row is the experiments on SVHN. The second row is the experiments on CIFAR-10. The last row is the experiments on CIFAR-100. The first column to the last column are the experiments of $\epsilon$ equal to 2, 4, 6, 8, and 10, respectively.

**Cyclic Learning Rate.** We illustrate the experiments of adversarial training using cyclic learning rate. The learning rate increases linearly in the first 80 epochs and decreases to zero in the last 120 epochs. This learning rate mainly contributes to optimization. In terms of generalization, as discussed in theoretical settings, the generalization bound is no larger than that of the previous cases.

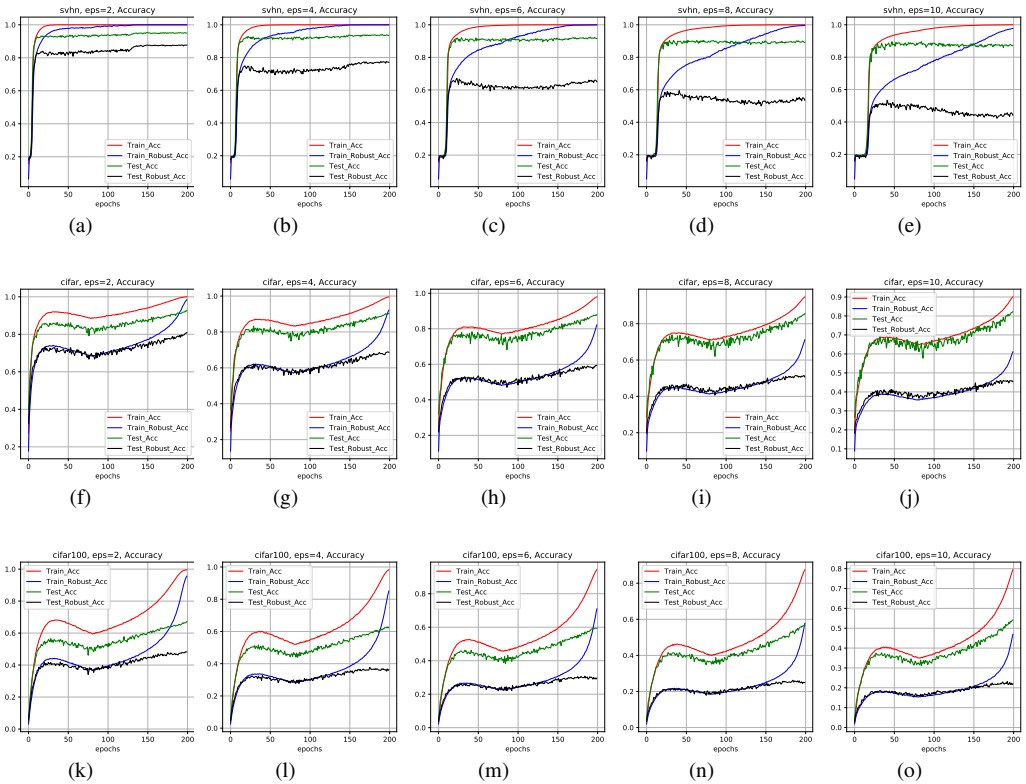

Figure 7: Accuracy of adversarial training with super-converge learning rate. The first row is the experiments on SVHN. The second row is the experiments on CIFAR-10. The last row is the experiments on CIFAR-100. The first column to the last column are the experiments of $\epsilon$ equal to 2, 4, 6, 8, and 10, respectively.