# OpenReview forum: "Stability Analysis and Generalization Bounds of Adversarial Training"
_NeurIPS.cc/2022/Conference — NeurIPS 2022 Accept_

### Official Review · Reviewer_4YYt · 2022-07-04

**Rating:** 7
**Confidence:** 4
**Soundness:** 3 good
**Presentation:** 3 good
**Contribution:** 3 good

**Summary:**

This paper studies the algorithmic stability of adversarial training. Compared to existing literature, it provides a better way to describe the smoothness of adversarial training, and obtain a more intuitive description on the algorithmic stability, convergence, and generalization error in adversarial training. Theorems and experiments are provided to justify their arguments.

**Questions:**

Please help address my concerns in the weaknesses section.

**Strengths And Weaknesses:**

Strengths:

[1] Different from existing literature, this paper provides another aspect of approximate smoothness to analyze the algorithmic stability of adversarial training. General results in algorithmic stability may not fit adversarial training very well, and this work overcomes this issue.

[2] This paper provides theoretical results in terms of algorithmic stability, generalization, and optimization, which together characterize the adversarial training process. The theorems are easy to understand. They also provide a lower bound result, which further strengthens their contribution.

Weaknesses:

[1] Given a real data set, the relationship between \eps and \eta is fixed. Therefore, from the current experiments, it is not so convincing that the large generalization gap is caused by a large \eta. It would be great if the authors can provide some simulation studies to verify the relationship between \eta and the generalization when varying L and \eps. It would be even better if there is a simulation in other tasks rather than adversarial training to justify the relationship between \eta and generalization error.

[2] This paper does not provide enough new insights on how to improve adversarial training. For example, this paper argues the necessity of early stopping, but it is a known conclusion as in Bassily et. al. (2020). In Xing et. al. (2021), they consider the noise-injection method to improve robustness, which is beyond Bassily et. al. (2020) and particular for adversarial training. However, in this submission, the authors only mention some potential solutions without any strong evidence.

[3] This submission compares a lot with Xing et. al. (2021), and mentions a lot of weaknesses of that paper. However, after reading Xing et. al. (2021), my understanding is that they directly verify the effectiveness of the noise injection (optimal rate), so there is no need to work on a fine-grained analysis on a sub-optimal rate in their paper. Please tune your description when describing Xing et. al. (2021).

---

> ### Author Response · Authors · 2022-08-02
> **Response to Reviewer 4YYt**
>
> We thank reviewers for the questions and comments. Below we answer your questions.
> ___
> **W1**: Given a real data set, the relationship between $\epsilon$ and $\eta$ is fixed ... It would be even better if there is a simulation in other tasks rather than adversarial training to justify the relationship between $\eta$ and generalization error.
>
> A: Thanks for the suggestion. Since our work focuses on adversarial generalization and robust overfitting, we have not considered other tasks. Based on your suggestion, we are trying to design simulations for other tasks to study the effects $\eta$.
> ___
> **W2**: This paper does not provide enough new insights on how to improve adversarial training. ...  in this submission, the authors only mention some potential solutions without any strong evidence.
>
> A: Thanks for the question. To improve training, there are two ways. The first one is to control the assumptions given in 4.1. The second one is to design new algorithms to improve the bounds based on the assumptions.
>
> For the first one, we have discussed one possible way, which is to control $l_z$. However, our work is a mainly theoretical work to provide generalization bound and provide an understanding of adversarial machine learning. To provide strong evidence to show that controlling $l_z$ does improve training beyond the scope of our work.
>
> As for algorithms, it is one of the weaknesses of the stability framework that it cannot provide new insight into design new algorithm. Notice that the stability-based generalization bound is algorithm-dependent. The researcher should propose an algorithm first. Then, they can analyze the stability of the proposed algorithm. This is similar to the optimization field. It is like 'providing the convergence rate of one algorithm cannot provide new insight to design new algorithm.'
>
> However, there is still a possible way for the researcher to design new algorithms. The researcher could dig into the proof of the bound rather than the bound itself to think about what algorithm can reduce the generalization bound. We will keep studying these in the future.
> ___
> **W3**: This submission compares a lot with Xing et. al. (2021), and mentions a lot of weaknesses of that paper.
>
> A: Thanks for the comments. We will modify it as follows to avoid misleading the readers' understanding of the work of [Xing et. al. (2021)].
>
> Comparison of our work and the work of [Xing et. al. (2021)]. The work of [Xing et. al. (2021)] argued that the max function is not smooth even though the standard counterpart is smooth. Therefore, they followed the bound in non-smooth cases [Bassily et. al. (2020)]. Then, the work of [Xing et. al. (2021)] aimed to solve the non-smooth issue. They design a noise-injected algorithm and show its effectiveness in tackling the non-smooth issue. Our work focus on providing better bounds to interpret robust overfitting.
>
> Please see our updated version (upload later).

---

> > ### Comment · Reviewer_4YYt · 2022-08-04
> > **Thanks for your response**
> >
> > I appreciate the authors addressing my comments. I have read the reviews and the author response, and raised my score from 6 to 7.
> >
> > Please notify me when the revision of the paper is updated. Thank you.

---

> > > ### Author Response · Authors · 2022-08-08
> > > **Response to Reviewer 4YYt**
> > >
> > > Thanks for the response. We have uploaded the updated paper.

---

### Official Review · Reviewer_zt8C · 2022-07-11

**Rating:** 7
**Confidence:** 4
**Soundness:** 3 good
**Presentation:** 4 excellent
**Contribution:** 4 excellent

**Summary:**

This paper provides a stability analysis ( and the consequent generalization analysis) of adversarial training. In order to tackle the nonsmoothness of adversarial loss, it introduces the eta approximation smoothness as an analysis tool. It derives a better rate than prior works with new theoretical insights.

**Questions:**

One suggestion is that the theory considers gradient descent of empirical adversarial loss, while in practice, the optimization is done by an alternative min-max type optimization. I wonder whether this gap is essential or not.

**Limitations:**

N.A.

**Strengths And Weaknesses:**

The paper is well written and I believe it makes a good contribution to this line of research.

---

> ### Author Response · Authors · 2022-08-02
> **Response to Reviewer zt8C**
>
> We thank reviewers for the questions and comments. Below we answer your questions.
>
> Q: One suggestion is that the theory considers gradient descent of empirical adversarial loss, ... I wonder whether this gap is essential or not.
>
> A: Thanks for the suggestion. Our Corollary 5.2 might answer your question. For the inner max problem, we assume that the algorithms (e.g., PGD) find suboptimal adversarial examples ($\Delta\epsilon$ from the optimal one). Our results show that the generalization bound of this type of min-max algorithm has the same order as that of running SGD on adversarial loss.

---

### Official Review · Reviewer_mocV · 2022-07-16

**Rating:** 7
**Confidence:** 3
**Soundness:** 3 good
**Presentation:** 4 excellent
**Contribution:** 3 good

**Summary:**

- The paper shows that the adversarial loss is approximately gradient Lipschitz, with the smoothness dependent on the adversarial perturbation size $\varepsilon$
- The paper derives an $\varepsilon$ dependent generalization bounds, and interprets the bounds to argue that overfitting is a result of the non-smoothness of the adversarial loss
- The paper analyzes common methods to mitigate adversarial overfitting to show that they are stability promoting
- Experiments on SVHN, CIFAR-10, CIFAR-100, Imagenet to verify the bounds

**Questions:**

1. See weaknesses 1, 2. Do the authors have opinions on this?
2. Does robust overfitting affect non-adversarial training based defenses such as randomized smoothing? Perhaps this could be useful evidence in support of the paper in case these other methods do face the same overfitting issue.

**Limitations:**

See weaknesses/questions section.

**Strengths And Weaknesses:**

Strengths:
1. Good presentation, and explains necessary context/background well
2. Interesting result, and the paper provides a clear interpretation of it to explaining the robust overfitting phenomenon. The derived bound seems significant and is potentially useful beyond adversarial training literature. Very clear explanation of how this result compares to existing bounds
3. Section 6 provides interpretations of methods which have been shown to reduce robust overfitting - this part is quite interesting

Weaknesses:
1. The derived result doesn't seem to explain why robust generalization requires significantly more datapoints (compared to regular training), and why this reduces robust overfitting. It seems that the bound derived would imply that robust overfitting would occur at the same rate regardless of $n$. We understand that this is perhaps outside of the scope of the paper, however a mention of this missing piece might be helpful
2. A limitations sections explaining aspects of robust overfitting remain unexplained by this bound would be useful.
3. Experiments feel a bit rudimentary, i.e. it feels like the experiments section verifies already known results, without a real attempt at nitpicking/ablating the theory. The reviewer is unsure what experiments could be conducted to improve this section

---

> ### Author Response · Authors · 2022-08-02
> **Response to Reviewer mocV**
>
> We thank reviewers for the questions and comments. Below we answer your questions.
> ___
> **Q1.1 (W1)**: The derived result doesn't seem to explain why robust generalization requires significantly more datapoints (compared to regular training), and why this reduces robust overfitting. It seems that the bound derived would imply that robust overfitting would occur at the same rate regardless of n.
>
> A: Thanks for the question. Our bound in Eq. (6.2) can provide an explanation for why more training data can reduce robust overfitting. The bound derived would imply that robust overfitting would occur at a different rate in terms of n.
>
>
> Firstly, the bound in Eq. 6.2 is
> $$\text{excess risk}\leq L\eta T\alpha +2L^2 T\alpha/n+D^2/T\alpha+L^2\alpha,$$
> where $n$ is the number of samples, $T$ is the number of epochs, and $\alpha$ is the step size. In the paper, we mainly discuss the regime in that the first term dominates the other term. However, the actual speed (rate of decrease) of robust overfitting is given in the first two term
> $$L(\eta+L/n)T\alpha.$$
> For larger $n$, the slope is smaller. Then, the decrease rate is smaller. It might (partially) answer your question.
> ___
> **Q1.2 (W2)**: A limitations sections explaining aspects of robust overfitting remain unexplained by this bound would be useful.
>
> A: Thanks for the suggestion. This is related to the last question. In our opinion, one thing remains unexplained is that whether the decrease rate is $O(T)$, $O(\sqrt{T})$, or in between. First of all, this is the gap between the upper and lower bounds. Secondly, we can see in the experiments that the decrease rate is closed to $O(T)$ when $\epsilon$ is small and is closed to $O(\sqrt{T})$ when $\epsilon$ is large. We will add a section to discuss the limitation. We will upload the updated version later.
> ___
> **W3**: Experiments feel a bit rudimentary, i.e., it feels like the experiments section verifies already known results, without a real attempt at nitpicking/ablating the theory. The reviewer is unsure what experiments could be conducted to improve this section.
>
> A: Thanks for the question. Our experiments make the following attempts to ablating the theory. Since the main difference between our bound and the existing bounds is the term $L_z\epsilon T\alpha$. Our ablation studies are to vary $\epsilon$ (=2,4,6,8, and 10) and to vary $T\alpha$ (we consider different learning rate schedules). In adversarial training, $L_z$ is unknown, and it is hard to provide experiments to do ablation studies with respect to $L_z$.
> ___
> **Q2**: Does robust overfitting affect non-adversarial training-based defenses such as randomized smoothing? Perhaps this could be useful evidence in support of the paper in case these other methods do face the same overfitting issue.
>
> A: As we can see in research in the last few years, non-AT defenses do face robust overfitting issues. However, using them as evidence in support of the paper requires providing a generalization bound of each of these algorithms independently.
>
> As we can see in the paper, the stability-based generalization bound is algorithm-dependent. For example, providing a generalization bound of AT (Thm 5.1) and SWA (Thm. 6.2) requires two different proofs.   It may be straightforward to provide stability-based generalization bound of some of the defense algorithms and may be highly nontrivial for some other algorithms. We will study these problems in the future.

---

### Official Review · Reviewer_LfVE · 2022-07-17

**Rating:** 7
**Confidence:** 4
**Soundness:** 3 good
**Presentation:** 3 good
**Contribution:** 3 good

**Summary:**

The paper is concerned with a uniform stability analysis of adversarial training, in order to gain a better understanding of the robust overfitting phenomenon.
To this end the authors introduce a class of $\eta$-approximately smooth functions, i.e., functions $h$ whose gradient is approximately Lipschitz continuous and satisfies
$$\|\nabla h(\theta_1) - \nabla h(\theta_2)\|\leq L \|\theta_1-\theta_2\| + \eta.$$
This is motivated by the fact that the adversarial loss $h(\theta; z) := \max_{\|z'-z\|\leq\varepsilon} g(\theta; z')$ is $\eta$-approximately smooth with $\eta=2 L_z \varepsilon$ where $L_z$ is the Lipschitz constant of $z\mapsto\nabla_\theta g(\theta, z)$.

The authors then prove bounds for the generalization and the optimization gap of stochastic gradient descent (SGD) applied to the adversarial risk for different assumptions on the loss function (such as different degrees of convexity). The most notable discovery is that the generalization gap after $T$ steps of SGD with learning rate $\alpha$ contains and extra term of order $L_z \varepsilon T \alpha$. The authors make the point that this explains robust overfitting since this term blows up for more and more SGD steps. This is also indicated by their numerical results.



**Questions:**

I have a couple of questions and remarks which should be taken into account for a revised version of the article:

- a short paragraph on notation would be good. In particular, you should remark that you use the Euclidean norm not any norm.
- Assumption 4.1: comment on the third assumption (see above)
- Lemma 4.2: the comments before and after the lemma are a little confusing: This lemma shows that the adversarial loss is $\eta$-approximately smooth and not that its not Lipschitz smooth. For this you would need a lower bound.
- Lemma 4.2: Why is $h$ differentiable with respect to $\theta$? A priori, this is not obvious due to the maximum in $z$.
- line 170: Calling (4.2) SUBgradient descent seems out of place. This is just stochastic gradient descent and the gradient is a proper gradient by your assumptions.
- line 198: did you mean to refer to Eq. (5.1) instead of (5.3)?
- Corollary 5.2 basically shows that the inner maximization problem does not play a role at all in your analysis and you could just choose an arbtrary point which is $\varepsilon$-close to $z$. I suspect this is due to the strong Lipschitz assumption on the gradient of $h$ (see above). This should be pointed out.
- Theorem 5.3 should read: "IT exists a function h...", "THERE EXIST data sets S and S'..."
- line 223: Is the improvement from L (the $\theta$ Lipschitz constant of the loss) to $L_z\varepsilon$ really an improvement?
- line 223: I thought $L$ is the Lipschitz constant of the loss and not its gradient?
- lines 223-230: the language of this paragraph is clumsy and should be revised.
- Theorems 5.4 and 5.5 have slightly different wording which is confusing. For instance, "we run SGD" versus "we run SGD on the adversarial surrogate". Are there different assumptions here?
- Theorem 5.4: It should read $\alpha\leq 1/\beta$.
- Maybe formula (6.2) could be moved to the beginning of the article. It seems to contain the main contribution.
- Could one use a discrepancy principle (as used for variational regularization schemes to solve inverse problems) for the early stopping?
- lines 347-348: For small values of $\varepsilon$ I agree that the robust test accuracy seems to behave linearly in $\varepsilon$ but not necessarily for large values ($\varepsilon=10$).

**Limitations:**

The limitations of the assumption of an input-Lipschitz continuous gradients of the loss function should be discussed.

Negative societal impact is not to be expected.

**Strengths And Weaknesses:**

Strenghts:

The paper is structured in a clear way and the arguments are easy to follow. The analysis of gradient descent type schemes for $\eta$-approximately smooth functions is a valuable contribution on its own (at least I think its novel). The main strengths of the contribution are (1) a uniform stability analysis which depends on the adversarial budget $\varepsilon$ and extends to non-convex losses and (2) a theoretical explanation of robust overfitting.

Weaknesses:

My main criticism is that the Lipschitz gradient assumption $\|\nabla_\theta g(\theta, z_1) - \nabla_\theta g(\theta,z_2)\| \leq L_z \|z_1 - z_2\|$ $\forall \theta$ with respect to the input variable $z$ seems very strong. Note that $L_z$ multiplies the adversarial budget $\varepsilon$ in the stability estimates. Standard network architectures are known to be very far from being Lipschitz continuous with a moderate constant. In the light of this, demaning uniform Lipschitz continuity of the loss w.r.t. $z$ appears to be questionable. This is particularly important since recently it was shown that adversarial training leads to a total variation regularization instead of Lipschitz regularization, see [1,2]. Hence, in practice such uniform Lipschitz continuity may get lost during the optimization.
This issue requires some more explanations and justifications.
Besides that, the paper would benefit from a careful language revision.

[1] Finlay, C., Oberman, A. M., & Abbasi, B. (2018). Improved robustness to adversarial examples using Lipschitz regularization of the loss.
[2] Bungert, L., Trillos, N. G., & Murray, R. (2021). The geometry of adversarial training in binary classification. arXiv preprint arXiv:2111.13613.

---

> ### Author Response · Authors · 2022-08-02
> **Response to Reviewer LfVE (2/2)**
>
> Q11: lines 223-230: the language of this paragraph is clumsy and should be revised.
>
> A: Thanks for the suggestion. We will revise (including Q9 - Q11) it in the updated version.
>
> Q12: Theorems 5.4 and 5.5 have slightly different wording, which is confusing. For instance, "we run SGD" versus "we run SGD on the adversarial surrogate". Are there different assumptions here?
>
> A: There are no different assumptions. We will use consistent descriptions in the updated version.
>
> Q13: Theorem 5.4: It should read $1/\beta$.
>
> A: Thm 5.4 uses a diminishing step size $1/(\beta t)$. Using fixed stepsize, we cannot obtain a bound in $O(T)$. We will emphasize the diminishing step size in the updated version.
>
> Q14: Maybe formula (6.2) could be moved to the beginning of the article. It seems to contain the main contribution.
>
> A: Thanks for the suggestion. We will use 6.2 to illustrate our main results and place them in Intro.
>
> Q15: Could one use a discrepancy principle (as used for variational regularization schemes to solve inverse problems) for the early stopping?
>
> A: After reading the work of [1,2], we think using total variation regularization schemes as stopping criteria for early stopping makes sense intuitively. However, without comprehensive experiments, we cannot have a sure answer.
>
> Q16: lines 347-348: For small values of, I agree that the robust test accuracy seems to behave linearly in but not necessarily for large values.
>
> A: Yes we agree. We will remove the word 'linealy'.

---

> ### Author Response · Authors · 2022-08-02
> **Response to Reviewer LfVE (1/2)**
>
> We thank Reviewer LfVE for the questions and comments. Especially, we would like to thank Reviewer LfVE for giving many constructive suggestions to improve our paper. We will upload our updated version later. Below we answer your questions.
> ___
> **Weakness (Q2)**: My main criticism is that the Lipschitz gradient assumption is strong ....This is particularly important since recently it was shown that adversarial training leads to a total variation regularization instead of Lipschitz regularization, see [1,2]. Hence, in practice such uniform Lipschitz continuity may get lost during the optimization.
>
> A: Thanks for the question. After reading the paper [1,2], we find the criticism should be function Lipschitz assumption is strong rather than the gradient Lipschitz assumption is strong. Therefore, it is not contradicting assumption 4.1 and is consistent with the goal of our paper in some sense (We aim to reduce the dependence on function Lipschitz since it is a strong assumption).
>
> To be specific, in the work of [1,2], the authors show that adversarial training is equivalent to minimizing a total variation regularization problem
>
> $$E _{(x,y)}[\ell(x,y)]+\epsilon E _{(x,y)}\|\nabla _x \ell(x,y)\| _*,$$
>
> where $\|\cdot\|  _*$ is the dual norm, instead of a Lipschitz regularization problem
>
> $$E _{(x,y)}[\ell(x,y)]+\epsilon \max _{(x,y)}\|\nabla _x \ell(x,y)\| _*.$$
>
> Notice that the Lipschitz regularization term is a penalty term on the magnitude of the gradient, which is equivalent to the penalty of the function Lipschitz rather than gradient Lipschitz.
> It is because $\|\nabla f(x)\|\leq L$ is equivalent to $|f(x)-f(y)|\leq L\|x-y\|$. Thus, the argument should be that 'uniform function Lipschitz with respect to the input $x$ may get lost during the optimization. '
>
> Therefore, it is not contradicting assumption 4.1 since we do not need to assume that $g$ is function Lipschitz with respect to $z$. It is consistent with the goal of our paper.
> ___
> Q1: a short paragraph on notation would be good. In particular, you should remark that you use the Euclidean norm not any norm.
>
> A: Thanks for the suggestion. We will add such a paragraph.
>
> Q3: Lemma 4.2: the comments before and after the lemma are a little confusing: This lemma shows that the adversarial loss is $\eta$-approximately smooth and not that its not Lipschitz smooth. For this you would need a lower bound.
>
> A: Thanks for pointing out the confusing comments. We will rewrite 'the adversarial loss is approximately smooth (define later in Def. 4.3).'
>
> Q4: Lemma 4.2: Why is $h$ differentiable? A priori, this is not obvious due to the maximum in $z$.
>
> A: In Lemma 4.2, $h$ is not guaranteed to be differentiable. Lemma 4.2 holds for all the subgradients in the subdifferential at point $\theta$. We will fix the description.
>
> Q5: line 170: Calling (4.2) SUBgradient descent seems out of place. This is just stochastic gradient descent, and the gradient is a proper gradient by your assumptions.
>
> A: In definition 4.3, we will modify 'We say a differentiable function.' In line 170, we will modify `stochastic gradient descent'. In our analysis, starting from Sec 4.3, we assume $h$ to be differentiable to simplify the argument.
>
> Q6: line 198: did you mean to refer to Eq. (5.1) instead of (5.3)?
>
> A: Yes, this is a typo. We will fix it.
>
> Q7: Corollary 5.2 basically shows that the inner maximization problem does not play a role at all in your analysis, and you could just choose an arbitrary point which is -close to. I suspect this is due to the strong Lipschitz assumption on the gradient of  (see above). This should be pointed out.
>
> A: Yes, we agree. Corollary 5.2 implies that the generalization bound of (weak attack) AT shares the same order as that of (strong attack) AT. This might be due to the assumption and/or the weakness of the stability framework. We will add a discussion. However, the exact bound (not in order) of (weak attack) AT is still larger, which still matches the observation in practice.
>
> Q8: Theorem 5.3 should read: "IT exists a function h...", "THERE EXIST data sets S and S'..."
>
> A: Thanks for pointing out the typo. We will fix it.
>
> Q9: line 223: Is the improvement from $L$ to $L_z\epsilon$ really an improvement?
>
> A: This question is related to the weakness part. See our responses above.
>
> Q10: line 223: I thought $L$ is the Lipschitz constant of the loss and not its gradient?
>
> A: Thanks for the question. For differentiable function, function Lipschitz $|f(\theta_1)-f(\theta_2)|\leq L|\theta_1-\theta_2|$ implies bounded gradient $\|\nabla f(\theta)\|\leq L$. We will state it clearly.

---

> > ### Comment · Reviewer_LfVE · 2022-08-08
> > **Response**
> >
> > Thanks for carefully addressing my comments!
> >
> > There is still a little confusion with respect to the Lipschitz gradient assumption that is also partially due to my own imprecision. Indeed, the works [1,2] show that adversarial training leads to a total variation regularization and no Lipschitz regularization wrt the input variable. In contrast, assumption 4.1 demands that the gradient wrt the parameters is Lipschitz continuout wrt to the input variable.
> >
> > Speaking in terms of derivatives, a function Lipschitz assumption (which is unrealistic and which the authors indeed do not pose) would be $|\nabla_z g(\theta,z)|\leq L$ for all $\theta$ and all $z$.
> > The assumption that the authors do pose is the gradient Lipschitz assumption $|\nabla_z \nabla_\theta g(\theta, z)|\leq L$ for all $\theta$ and all $z$. This looks equally strong or even stronger to me and in the revised version I would appreciate an example or a comment addressing whether this is a realistic assumption. For instance, for feedforward neural networks with Lipschitz continuous activation a uniform bound on the product of the weight matrices ensures a function Lipschitz assumption. Does this also imply a gradient Lipschitz assumption or does one possibly need a condition on the data set, as well?

---

> > > ### Author Response · Authors · 2022-08-08
> > > **Response to Reviewer LfVE**
> > >
> > > Thanks for the question.
> > >
> > > Firstly, for neural networks, $L_z$-gradient Lipschitz with respect to $z$ is a strong assumption due to the non-smooth activation function (e.g., ReLU). This is already discussed in the submitted version:
> > >
> > > ''While ReLU activation function is non-smooth, recent works [1,13] showed that the loss function of over-parameterized DNNs is
> > > semi-smooth.''
> > >
> > > We use the work of [1,13] to argue that neural networks is closed to a smooth function, $L_z$-gradient Lipschitz is an acceptable assumptions.
> > >
> > > Secondly, for neural networks with smooth activation function (e.g., SiLU), one can estimate a uniform bound of gradient Lipschitz of DNNs using the same Lipschitz-peeling technique as you mentioned.
> > >
> > > Lastly, gradient Lipschitz is the most commonly used assumption in both optimization and generalization fields. Without this assumption, it is hard to provide an analysis of an algorithm.

---

### Author Response · Authors · 2022-08-08
**Summary of Revision**

Dear AC and Reviewers,

We have provided the updated version of our paper. We fix the typos and inconsistent statement mentioned by the reviewers.

Sec.1: We rewrite the description of the work of [46]. We move Eq. (6.2) to Intro to describe the main result of our paper.

Sec. 4: We use subgradient in Lemma 4.1.

Sec. 5: We add a remark to state the limitation of Corollary 5.1. We rewrite the comparison with existing bounds in Sec 5.2.

Sec. 8: We add a discussion on limitation and future work.

Sec. A.2: We rewrite the proof in terms of sub-gradient.

---

### Meta-Review · Area_Chair_RtE3 · 2022-08-29

**Recommendation:** Accept
**Confidence:** Certain

**Metareview:**

This paper uses uniform stability to study generalization in networks under adversarial training. The authors use their framework to understand robust overfitting: the phenomenon in which a network is adversarially robust on the training set, but generalize poorly. The theoretical results in this paper are backed up by experiments and applications to provide theoretical clarity to a number of common methods for reducing overfitting.

All reviewers supported acceptance of this paper. Reviewers enjoyed the exposition and writing, and found the theoretical developments and explanation of overfitting to be convincing. Finally, the reviewers liked the experiments backing up the theory and the practical applications.


**Award:**

No

---

### Decision · Program_Chairs · 2022-09-14

Accept